# MAG-GNN: Reinforcement Learning Boosted Graph Neural Network

**Lecheng Kong**[1]  **Jiarui Feng**[1]  **Hao Liu**[1]  **Dacheng Tao**[2]
**Yixin Chen**[1]  **Muhan Zhang**[3]
{jerry.kong, feng.jiarui, liuhao, ychen25}@wustl.edu,
dacheng.tao@gmail.com, muhan@pku.edu.cn
[1]Washington University in St. Louis  [2]JD Explore Academy  [3]Peking University

## Abstract

While Graph Neural Networks (GNNs) recently became powerful tools in graph learning tasks, considerable efforts have been spent on improving GNNs' structural encoding ability. A particular line of work proposed subgraph GNNs that use subgraph information to improve GNNs' expressivity and achieved great success. However, such effectivity sacrifices the efficiency of GNNs by enumerating all possible subgraphs. In this paper, we analyze the necessity of complete subgraph enumeration and show that a model can achieve a comparable level of expressivity by considering a small subset of the subgraphs. We then formulate the identification of the optimal subset as a combinatorial optimization problem and propose Magnetic Graph Neural Network (MAG-GNN), a reinforcement learning (RL) boosted GNN, to solve the problem. Starting with a candidate subgraph set, MAG-GNN employs an RL agent to iteratively update the subgraphs to locate the most expressive set for prediction. This reduces the exponential complexity of subgraph enumeration to the constant complexity of a subgraph search algorithm while keeping good expressivity. We conduct extensive experiments on many datasets, showing that MAG-GNN achieves competitive performance to state-of-the-art methods and even outperforms many subgraph GNNs. We also demonstrate that MAG-GNN effectively reduces the running time of subgraph GNNs.

## 1 Introduction

Recent advances in Graph Neural Networks (GNNs) greatly assist the rapid development of many areas, including drug discovery [2], recommender systems [31], and autonomous driving [6]. The power of GNNs has primarily been attributed to their Message-Passing Paradigm [12]. Message-Passing Paradigm simulates a 1-dimensional Weisfeiler-Lehman (1-WL) algorithm for graph isomorphism testing. Such a simulation allows GNN to encode rich structural information. In many fields, structural information is crucial to determine the properties of a graph.

However, as Xu *et al.* [32] pointed out, GNN's structure encoding capability, or its expressivity, is also upper-bounded by the 1-WL test. Specifically, a message-passing neural network (MPNN) cannot recognize many substructures like cycles and paths and fails to properly learn and distinguish regular graphs. Meanwhile, these substructures are significant in areas including chemistry and biology. To overcome this limitation, considerable effort was spent on investigating more-expressive GNNs. A famous line of work is *subgraph GNNs* [33, 34, 37]. Subgraph GNNs extract rooted subgraphs around every node in the graph and apply MPNN onto the subgraphs to obtain subgraph representations. The subgraph representations are summarized to form the final representation of the graph. Such an approach is theoretically proved to be more expressive than MPNN and achieved superior empirical results. Later work found that subgraph GNNs are still bounded by the 3-dimensional WL test (3-WL) [10].

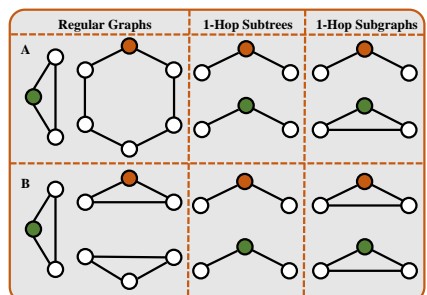

Figure 1: Comparison of two simple graphs.

This spurs the research on extensions to edge-rooted subgraph GNNs with better expressivity [15]. In fact, by linking subgraph GNNs to the $k$-dimensional WL ($k$-WL) test hierarchy, we observe that for subgraph GNNs to exceed the limits of k-WL expressivity, it needs to enumerate and apply MPNN to subgraphs exponential to $k$ [26, 10]. Hence, the higher expressivity, unfortunately, comes with a higher computational cost, which makes it challenging to apply subgraph GNNs to even moderately sized graphs (e.g., $\sim$100 nodes).

This makes us wonder, is it necessary for subgraph GNNs to enumerate all rooted subgraphs to achieve higher expressivity? For example, an MPNN fails to distinguish graphs A and B in Figure 1, as they are 2-regular graphs with identical 1-hop subtrees. Meanwhile, a subgraph GNN will see different subgraphs around nodes in the two graphs. These subgraphs are distinguishable by MPNN, allowing a subgraph GNN to differentiate between graphs A and B. However, we can observe that many subgraphs from the same graph are identical. Specifically, graph A has two types of subgraphs, while graph B only has triangle subgraphs. As a result, locating a non-triangle subgraph in the top graph enables us to run MPNN once on it to discern the difference between the two graphs. On the contrary, a subgraph GNN takes eight extra MPNN runs for the remaining nodes. This graph pair shows that we can obtain discriminating power equal to that of a subgraph GNN without enumerating all subgraphs. We also include advanced examples with more complex structures in Section 3.

Therefore, we propose Magnetic Graph Neural Network (MAG-GNN), a reinforcement learning (RL) based method, to leverage this property and locate the discriminative subgraphs effectively. Specifically, we start with a candidate set of subgraphs randomly selected from all rooted subgraphs. The root node features of each subgraph are initialized uniquely. In every step, each target subgraph in the candidate set is substituted by a new subgraph with more distinguishing power. MAG-GNN achieves this by mapping each target subgraph to a Q-Table, representing the expected reward of replacing the target subgraph with another potential subgraph. It then selects the subgraph that maximizes the reward. MAG-GNN repeats the process until it identifies the set of subgraphs with the highest distinguishing power. The resulting subgraph set is then passed to a prediction GNN for downstream tasks. MAG-GNN reduces subgraph GNN's exponentially complex enumeration procedure to an RL searching process with constant steps. This potently constrains the computational cost while keeping the expressivity.

We conduct extensive experiments on synthetic and real-world graph datasets and show that MAG-GNN achieves competitive performance to state-of-the-art (SOTA) methods and even outperforms subgraph GNNs on many datasets with a shorter runtime. Our work shows that partial subgraph information is sufficient for good expressivity, and MAG-GNN smartly locates expressive subgraphs and achieves the same goal with better efficiency.

## 2  Preliminaries

A graph can be represented as $G = \{V, E, X\}$, where $V$ is the set of nodes and $E \subseteq V \times V$ is the set of edges. Let $V(G)$ and $E(G)$ represent the node and edge sets of $G$, respectively. Nodes are associated with features $X = \{\boldsymbol{x}_v | \forall v \in V\}$. An MPNN $g$ can be decomposed into $T$ layers of COMBINE and AGGREGATE functions. Each layer uses the COMBINE function to update the current node embedding from its previous embedding and the AGGREGATE function to process the node's neighbor embeddings. Formally,

$$\boldsymbol{m}_v^{(t)} = \text{AGGREGATE}^{(t)}(\{\{\boldsymbol{h}_u^{(t-1)}, u \in \mathcal{N}(v)\}\}), \quad \boldsymbol{h}_v^{(t)} = \text{COMBINE}^{(t)}(\boldsymbol{m}_v^{(t)}, \boldsymbol{h}_v^{(t-1)}) \quad (1)$$

where $\boldsymbol{h}_v^{(t)}$ is the node representation after $t$ iterations, $\boldsymbol{h}_v^{(0)} = \boldsymbol{x}_v$, $\mathcal{N}(v)$ is the set of direct neighbors of $v$, $\boldsymbol{m}_u^{(t)}$ is the message embedding. $\boldsymbol{h}_v^{(T)}$ is used to form node, edge, and graph-level representations. We use $H = g(G)$ to denote the generated node embeddings of MPNN. MPNN's variants differ mainly by their AGGREGATE and COMBINE functions but are all bounded by 1-WL in expressivity.

This paper adopts the following formulation for subgraph GNNs. For a graph $G$, a subgraph GNN first enumerates all $k$-order node tuples $\{\boldsymbol{v} | \boldsymbol{v} \in V^k(G)\}$ and creates $|V^k(G)|$ copies of the graph. A

graph associated with node tuple $\boldsymbol{v}$ is represented by $G(\boldsymbol{v})$. Node-rooted subgraph GNNs adopt a 1-order policy and have $O(V(G))$ graphs; edge-rooted subgraph GNNs adopt a 2-order policy and have $O(V^2(G))$ graphs. Note that we are not taking exact subgraphs here, so we need to mark the node tuples on the copied graphs to maintain the subgraph effect. Specifically,

$$[X(\boldsymbol{v})]_{l,p} = \begin{cases} c^+ & \text{if } v_l = [\boldsymbol{v}]_j \text{ and } p = j \\ c^- & \text{otherwise} \end{cases} \quad \boldsymbol{v} \in V^k(G), \tag{2}$$

$X(\boldsymbol{v}) \in \mathbb{R}^{|V| \times k}$ and $G(\boldsymbol{v}) = \{V, E, X \oplus X(\boldsymbol{v})\}$, where $\oplus$ means row-wise concatenation. We use square brackets to index into a sequence. All entries in $X(\boldsymbol{v})$ are labeled as $c^-$ except those appearing at corresponding positions of the node tuple. An MPNN $g$ is applied to every graph, and we use a pooling function to obtain the collective embedding of the graph:

$$f_s(G) = R^{(P)}(\{g(G(\boldsymbol{v})) | \forall \boldsymbol{v} \in V^k(G)\}), \quad P \in \{G, N\}. \tag{3}$$

$f_s(G)$ can be a vector of graph representation if $R^{(P)}$ is a graph-level pooling function ($P$ equals $G$) or a matrix of node representations if $R^{(P)}$ is node-level ($P$ equals $N$). This essentially implements $k$-dimensional ordered subgraph GNN ($k$-OSAN) defined in [26] and captures most of the popular subgraph GNNs, including NGNN[35] and ID-GNN[33]. Furthermore, the expressivity of subgraph GNNs increases with larger values of $k$. Since node tuples, node-marked graphs, and subgraphs refer to the same thing, we use these terms interchangeably.

**The Weisfeiler-Lehman hierarchy (WL hierarchy).** The $k$-dimensional Weisfeiler-Lehman algorithm is vital in graph isomorphism testing. Earlier work established the hierarchy where the $(k+1)$-WL test is more expressive than the $k$-WL test [4]. Xu *et al.* [32] and Morris *et al.* [23] connected GNN expressivity to the WL test and proved that MPNN is bounded by the 1-WL test. Later work discovered that all node-rooted subgraphs are bounded by 3-WL expressivity [10], which cannot identify strongly regular graphs and structures like 4-cycles. Qian *et al.* [26] introduced $k$-dimensional ordered-subgraph WL ($k$-OSWL) hierarchy which is comparable to the $k$-WL test.

**Deep Q-learning (DQN).** DQN [19] is a robust RL framework that uses a deep neural network to approximate the Q-values, representing the expected rewards for a specific action in a given state. Accumulating sufficient experience with the environment, DQN can make decisions in intricate state spaces. For a detailed introduction to DQN, please refer to Appendix B.

## 3 Motivation

Figure 1 shows that while an MPNN encodes rooted subtrees, a subgraph GNN encodes rooted subgraphs around each node. This allows subgraph GNN to differentiate more graphs at the cost of $O(|V|)$ MPNN runs. Meanwhile, subgraph GNNs are still bounded by 3-WL tests. Hence, we may need at least 2-node-tuple-rooted (e.g., edge-rooted) subgraph GNNs, requiring $O(|V^2|)$ MPNN runs to obtain better expressivity. In fact, the required complexity of subgraph GNNs to break beyond $k$-WL expressivity grows exponentially with $k$. The high computational cost of high expressivity models prevents them from being widely applied to real-world datasets. A natural question is, *can we consider only a small subset of all the subgraphs to obtain similar expressivity*, just like in Figure 1, where one subgraph is as powerful as the collective information of all subgraphs? This leads us to the following experiment.

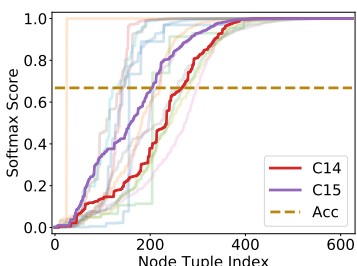

Figure 2: Sorted scores.

We focus on the SR25 dataset. It contains 15 different strongly-regular graphs with the same configuration, each of 25 nodes. The goal is to do multi-class classification to distinguish all pairs of graphs. Since node-rooted subgraph GNNs are upper-bounded by 3-WL and 3-WL cannot distinguish any strongly regular graphs with the same configuration, node-rooted subgraph GNNs will generate identical representations for the 15 graphs while performing 25 MPNN runs each. 2-node-tuple subgraph GNNs have expressivity beyond 3-WL and can distinguish any pair of graphs from the dataset, but it takes 625 MPNN runs.

To test if every subgraph is required, we train an MPNN on *randomly sampled* 2-node-marked graphs to minimize the expected loss to label $y$ of the unmarked graph $G$,

$$\min_{g_p} \mathbb{E}_{\boldsymbol{v}}[\mathcal{L}(\text{MLP}(f_r(G, \boldsymbol{v})), y)], \quad f_r(G, \boldsymbol{v}) = R^{(G)}(g_p(G(\boldsymbol{v}))), \quad \boldsymbol{v} \in V^2(G), \tag{4}$$

where $\mathcal{L}$ is the loss function, MLP is a multi-layer perceptron, $g_p$ is an MPNN, and $R^{(G)}$ pools the node representations to graph representations. Unlike 2-node-tuple-rooted subgraph GNNs that run the MPNN $|V^2|$ times for each graph, this model runs the MPNN exactly once. During testing, for each of the 15 graphs, we randomly sample one 2-node-marked graph for classification. We perform ten independent tests, and the average test accuracy is 66.8%. Using 2-node-marked graphs, with only one GNN run, it already outperforms node-rooted subgraph GNNs that fail on this dataset. More interestingly, for each graph $G$, we can sort the classification score of its $|V^2|$ possible node-marked graphs and plot them in Figure 2 (C14 and C15 are the plots for 14-th and 15-th graphs in the dataset). Note that the horizontal axis is not the number of subgraphs; it is the index of subgraphs after sorting by their *individual* classification scores. We see that each original graph has many marked graphs with a classification score close to one. That means even in one of the most difficult datasets, we still can find particular node-marked graphs that uniquely distinguish the original graph from others. Moreover, unlike the example in Figure 1 with only two types of subgraphs, these marked graphs fall into many different isomorphism groups, meaning that the same observation holds in more complex graphs and can be applied to a wide range of graph classes. We prove that such a phenomenon exists in most regular graphs, which cannot be distinguished by MPNNs (proof in Appendix A).

**Theorem 1.** *Let $G_1$ and $G_2$ be two graphs uniformly sampled from all $n$-node, $r$-regular graphs where $3 \leq r < \sqrt{2 \log n}$. Given an injective pooling function $R^{(G)}$ and an MPNN $g_p$ of 1-WL expressivity, with a probability of at least $1 - o(1)$, there exists a node (tuple) $\boldsymbol{v} \in V(G)$ whose corresponding node marked graph's embedding, $f_r(G_1, \boldsymbol{v})$, is different from any node marked graph's embedding in $G_2$.*

These observations show that by finding discriminative subgraphs effectively, we only need to apply MPNN to a much smaller subset of the large complete subgraph set to get a close level of expressivity.

## 4 Magnetic graph neural network

We formulate the problem of finding the most discriminative subgraphs as a combinatorial optimization problem. Given a budget of $m$ as the number of subgraphs, $k$ as the order of node tuples, and $g_p$ an MPNN that embeds individual subgraphs, we minimize the following individual loss to graph $G$,

$$\min_{U=(\boldsymbol{v}_1,...,\boldsymbol{v}_m) \in (V^k(G))^m} \mathcal{L}(\text{MLP}(f_p(G, U)), \boldsymbol{y})$$
$$f_p(G, U) = R^{(P)}(\{g_p(G(\boldsymbol{v})) | \forall \boldsymbol{v} \in U\}), \tag{5}$$

Note that this formulation resembles that of subgraph GNNs in Equation (3); we are only substituting $V^k$ with $U$ to reduce the number of MPNN runs. Witnessing the great success of deep RL in combinatorial optimization, we adopt Deep Q-learning (DQN) to our setting. We introduce each component of our DQN framework as follows.

**State Space:** For graph $G$, a state is $m = |U|$ node tuples, their corresponding node-marked graphs, and a $m$-by-$w$ matrix $W$ to record the state of $m$ node tuples. Our framework should generalize to arbitrary graphs. Hence, a state $s$ is defined as,

$$s = (G, U, W) = (G, (\boldsymbol{v}_1, ..., \boldsymbol{v}_m), W), s \in S = \mathcal{G} \times (\mathcal{V}^k)^m \times (\mathbb{R}^{m \times w}) \tag{6}$$

$S$ is the state space, $\mathcal{G}$ is the set of all graphs, and $\mathcal{V}^k$ is the set of all possible $k$ node tuples of $\mathcal{G}$. To generate the initial state during training, we sample one graph $G$ from the training set and randomly sample $m$ node tuples from $V^k(G)$. The state matrix $W$ is initialized to $\boldsymbol{0}$. The expressivity grows as $k$ grows. Generally, MAG-GNN with larger $k$ produces more unique graph embeddings, which is harder to train but might require smaller $m$ and fewer RL steps to represent the graph, leading to better inference time. However, for some datasets, such expressivity is excessive and poses great challenges to training. A smaller $k$ can reduce the sample space and stabilize training in this case.

**Action Space:** We define one RL agent action as selecting one index from one node tuple and replacing the node on that index with another node in the graph. This replaces the node-marked

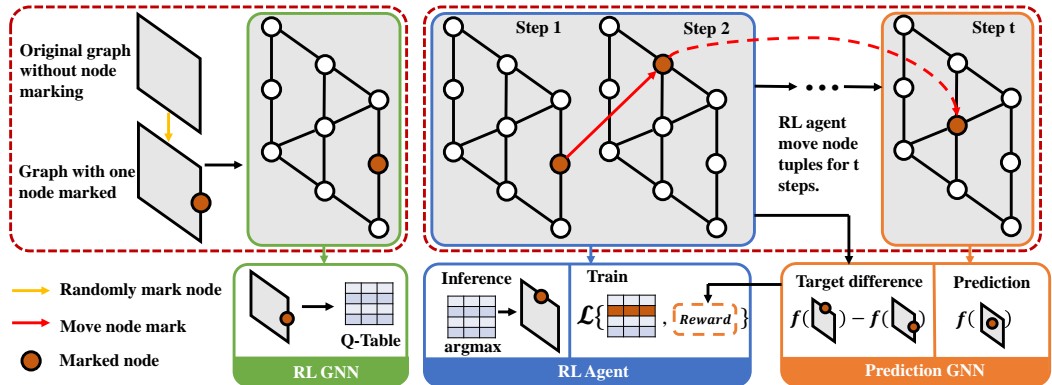

Figure 3: MAG-GNN's pipeline. An RL agent iteratively updates node tuples for better expressivity.

graph corresponding to the original node tuple with the one corresponding to the modified node tuple. Specifically, an action $a_{i,j,l}$ on state $s = (G, U, W)$ does the following on $U$:

$$U' = a_{i,j,l}(U) = (\boldsymbol{v}_1..., \boldsymbol{v}_{i-1}, \boldsymbol{v}_i', \boldsymbol{v}_{i+1}, ...\boldsymbol{v}_m), \boldsymbol{v}_i' = ([\boldsymbol{v}_i]_1, ..., [\boldsymbol{v}_i]_{j-1}, v_l, [\boldsymbol{v}_i]_{j+1}, ...[\boldsymbol{v}_i]_k) \quad (7)$$

The agent selects a target node tuple $\boldsymbol{v}_i$, whose $j$-th node is replaced with node $v_l \in V$. $W$ is then updated by an arbitrary state update function $W' = f_W(s, U')$ depending on the old state and new node tuples. The update function $f_W$ is not necessarily trainable (e.g., It can simply be a pooling function on embeddings of the marked nodes over states). The next state is $s' = (G, U', W')$. The action space is then $A = [m] \times [k] \times \mathcal{V}$. Actions only change the node tuple while keeping the graph structure, and the state matrix serves as a tracker of past states and actions. Unlike stochastic RL systems, our RL actions have deterministic outcomes.

The intuition behind the design of the action space is that it limits the number of actions for each node tuple to $O(m|V|k)$, which is linear in the number of nodes, and $k$ is usually small ($k = 2$ is already more expressive than most subgraph GNNs). We can further reduce the action space to $O(|V|k)$ if the agent does not need to select the update target but uses a Q-network to do Equation (7) on a given $\boldsymbol{v}_i$. In such a case, we either sequentially or simultaneously update all node tuples. Since the agent learns to be stationary when a node tuple should not be updated, we do not lose the power of our agent by the reduction. The overall action space design allows efficient computation of Q-values. We include a detailed discussion on the action space and alternative designs in Appendix C.1.

**Reward:** In Section 3, we show that a proper node-marked graph significantly boosts the expressivity. Hence, an optimal reward choice is the increased expressivity from the action. However, expressivity is itself vaguely defined, and we can hardly quantify it. Instead, since the relevant improvement in expressivity should affect the objective value, we choose the objective value improvement as the instant reward. Specifically, let $s = (G, U, W)$ be the current state and let $s' = (G, U', W') = a(s)$ be the outcome state of action $a$, the reward $r$ is

$$r(s, a, s') = \mathcal{L}(\mathrm{MLP}(f_p(G, U)), \boldsymbol{y}) - \mathcal{L}(\mathrm{MLP}(f_p(G, U')), \boldsymbol{y}) \quad (8)$$

This reward correlates the action directly with the objective function, allowing our RL agent to be task-dependent and more flexible for different levels of tasks.

**Q-Network:** Because our state includes graphs, we require an equivariant Q-network to output consistent Q-tables for actions. Hence, we choose MPNN to parameterize the Q-network. Specifically, for current state $s = (G, U, W)$ and the target node tuple $\boldsymbol{v} \in U$, we have the Q-table as,

$$[Q(s, \boldsymbol{v})]_l = \mathrm{MLP}([g_{rl}(G(\boldsymbol{v}))]_l \oplus \sum_{\boldsymbol{v} \in U} R^{(G)}(g_{rl}(G(\boldsymbol{v}))) \oplus R^{(W)}(W)) \quad (9)$$

Row $l$ in the Q-table is computed by the embedding of node $v_l$ in the node-marked graph by an MPNN $g_{rl}$, the current overall graph representation across all node tuples, and the state matrix $W$ summarized by a pooling function $R^{(W)}$. $[Q]_{l,j}$ represents the expected reward of replacing the node on index $j$ of node tuple $\boldsymbol{v}$ with node $v_l$. The best action $a_{j,l}$ is then chosen by,

$$\arg\max_{j,l}[Q(s, \boldsymbol{v})]_{l,j} \quad (10)$$

Note that because we assign different initial embeddings based on the node tuple, the MPNN distinguishes otherwise indistinguishable graphs.

As demonstrated in Figure 3, our agent starts with a set of random node tuples and their corresponding subgraphs. In each step, the agent uses an MPNN-parameterized Q-network to update one slot in one node tuple such that the new node tuple set results in a better objective value. The agent repeats for a fixed number of steps $t$ to find discriminative subgraphs. We do not assign a terminal state during training. Instead, the Q-Network will learn to be stationary when all other action decreases the objective value. This process is like throwing iron balls (marked nodes) into a magnetic field (geometry of the graph) and computing the force that the balls receive along with the interactions among balls (Q-network). We learn to move the balls and reason about the magnetic field's properties. Hence, we dub our method Magnetic GNN (MAG-GNN). To show the effectiveness of our method in locating discriminative node tuples, we prove the following theorem (proof in Appendix A).

**Theorem 2.** *There is a MAG-GNN whose action is more powerful in identifying discriminative node tuples than random actions.*

MAG-GNN is at least as powerful as random actions since we can adopt a uniform-output MPNN for the MAG-GNN, yielding random actions. The superiority proof identifies cases where MAG-GNN requires fewer steps to locate the discriminative node tuples. The overall inference complexity is the number of MPNN runs, $O(mtT|V^2|)$. A more detailed complexity analysis is in Appendix D.

Some previous works also realize the complexity limitation of subgraph GNNs and propose sampling-based methods, and we discuss their relationship to MAG-GNN. PF-GNN [7] uses particle-filtering to sample from canonical labeling tree. MAG-GNN and PF-GNN do not overlap exactly. However, we show that using the same resampling process, MAG-GNN captures PF-GNN (Appendix A).

**Theorem 3.** *MAG-GNN captures PF-GNN using the same resampling method in PF-GNN.*

k-OSAN [26] proposes a data-driven subgraph sampling strategy to find informative subgraphs by another MPNN. This strategy reduces to random sampling when the graph requires higher expressivity (e.g., regular graphs) and has no features because the MPNN will generate the same embedding for all nodes and hence cannot identify subgraphs that most benefit the prediction like MAG-GNN can. MAG-GNN does not solely depend on the data and finds more expressive subgraphs even without node features. Moreover, sampled subgraphs in previous methods are essentially independent. In contrast, MAG-GNN also models their correlation using RL. This allows MAG-GNN to obtain better expressivity with fewer samples and better consistency (More discussions in Appendix C.3).

### 4.1 Training MAG-GNN

With the state and action space, reward, and Q-network defined, we can use any DQN techniques to train the RL agent. However, to evaluate the framework's capability, we select the basic Experience Replay method [19] to train the Q-network. MAG-GNN essentially consists of two systems, an RL agent and a prediction MPNN. Making them coordinate is more critical to the method. The most natural way to train our system is first to train $g_p$, as introduced in Section 3 with random node tuples. We then use $g_p$ as part of $f_p$, the marked-graphs encoder, and treat $f_p$ as the fixed environment to train our Q-network. The advantage of the paradigm is that the environment is stable. Consequently, all experiences stored in the memory have the correct reward value for the action. This encourages stability and fast convergence during RL training. We term this paradigm ORD for ordered training.

However, not all $g_p$ trained from random node tuples are *good*. When we train $g_p$ for the ZINC dataset and evaluate all node-marked graphs to as in Section 3. The average standard deviation among all scores of node-marked graphs is ~0.003, and the average difference between the worst and best score is ~0.007. Hence, the maximal improvement is minimal if we use this MPNN as the environment. Intuitively, when the graph has informative initial features, $X$, like those in the ZINC dataset, the MPNN quickly recognizes patterns from these features while gradually learning to ignore node marking features $X(\boldsymbol{v}_i)$, as not all node markings carry helpful information. In such cases, we need to adjust $g_p$ while the RL agents become more powerful in finding discriminative subgraphs.

One way is to train the RL agent and $g_p$ simultaneously. Concretely, we sample a state $s$, run the RL agent $t$ steps to update it to state $s^t$, and train $g_p$ on the marked graphs of $s^t$. Then, in the same step, the RL agent samples a different state and treats the current $g_p$ as the environment to generate experience. Lastly, the RL agent is optimized on the sampled previous experiences. Because we adjust $g_p$ to capture node tuple information better, the score standard deviation of the node-marked ZINC graphs is kept at ~0.036. We term this paradigm SIMUL. Compared to ORD, SIMUL makes

the RL agent more difficult to train when $g_p$ evolves rapidly. Nevertheless, we observe that as $g_p$ gradually becomes stable, the RL agent can still learn effective policies.

One of the critical goals of MAG-GNN is to identify complex structural information that MPNN cannot. Hence, instead of training the agent on real-world datasets from scratch, we can transfer the knowledge from synthetic expressivity data to real-world data. As mentioned above, training MAG-GNN on graphs with features is difficult. Alternatively, we first use the ORD paradigm to train the agent on expressivity data without node features such as SR25. On these datasets, $g_p$ relies on the node markings to make correct predictions. We then fix the RL agent and only use the output state from the agent to train a new $g_p$ for the actual tasks, such as ZINC graph regression. Using this paradigm, we only need to train one MAG-GNN with good expressivity and adapt it to different tasks without worrying about overfitting and the aforementioned stability issue, we name this paradigm PRE. We experimented with different schemes in Section 6.

## 5    Related work

**More expressive GNNs.** A substantial amount of work strive to improve the expressivity of GNNs. They can be classified into the following categories: (1) Just like MPNN simulates the 1-WL test, **Higher-order GNNs** design GNNs to simulate higher-order WL tests. They include k-GNN [18], RingGNN [5], and PPGN [20]. These methods perform message passing on node tuples and have complexity that grows exponentially with $k$ and hence does not scale well to large graphs. (2) Realizing the symmetry-breaking power of subgraphs, **Subgraph GNNs**, including NGNN [35], GNN-AK [37], KPGNN [8], and ID-GNN [33], use MPNNs to encode subgraphs instead of subtrees around graph nodes. Later works, like I$^2$-GNN [15], further use 2-order(edge) subgraph information to improve the expressivity of subgraph GNNs beyond 3-WL. Recent works, such as OSAN [26] and SUN [10], unify subgraph GNNs into the WL hierarchy, showing improvement in expressivity for subgraph GNNs also requires exponentially more subgraphs. (3) **Substrcuture counting** methods, including GSN [3] and MotifNet [22], employ substructure counting in GNNs. They count predefined substructures undetectable by the 1-WL test as features for MPNN to break its expressivity limit. However, the design of the relevant substructures usually requires human expertise. (4) Many previous works also realize the complexity issue of more expressive GNNs and strive to reduce it. SetGNN [38] proposes to reduce node tuple to set and thus reduce the number of nodes in message passing. GDGNN [16] designs geodesic pooling functions to have strong expressivity without running MPNN multiple times. (5) **Non-equivariant GNNs.** Abboud *et al.* [1] proves the universality of MPNN with randomly initialized node features, but due to the large search space, such expressivity is hardly achieved. DropGNN [24] randomly drops out nodes from the graph to break symmetries in graphs. PF-GNN [7] implements a neural version of canonical labeling and uses particle filtering to sample branches in the labeling process. OSAN [26] proposes to use input features to select important subgraphs. Agent-based GNN [21] initializes multiple agents on a graph without the message-passing paradigm to iteratively update node embeddings. MAG-GNN also falls into this category.

**Reinforcement Learning and GNN.** There has been a considerable effort to combine RL and GNN. Most work is on application. GraphNAS [11] uses GNN to encode neural architecture and use reinforcement learning to search for the best network. Wang *et al.* [30] uses GNN to model circuits and RL to adjust the transistor parameters. On graph learning, most works use RL to optimize particular parameters in GNN. For example, SUGAR [27] uses Q-learning to learn the best top-k subgraphs for aggregations; Policy-GNN [17] learns to pick the best number of hops to aggregate node features. GPA [13] uses Deep Q-learning to locate the valuable nodes in active search. These works leverage the node feature to improve the empirical performance but fail to identify graphs with symmetries, while MAG-GNN has good expressivity without node features. To the best of our knowledge, our work is the first to apply reinforcement learning to the graph expressivity problem.

## 6    Experimental results

In the experiment [1], we answer the following questions: **Q1**: Does MAG-GNN have good expressivity, and is the RL agent output more expressive than random ones? **Q2**: MAG-GNN is graph-level;

---

[1]The code can be found at https://github.com/LechengKong/MAG-GNN

| Table 1: Synthetic results. (↑) | | | |
|---|---|---|---|
| | EXP | CSL | SR25 |
| GIN [32] | 50.0 | 10.0 | 6.7 |
| RNI [1] | 99.7 | 16.0 | 6.7 |
| NGNN [35] | 100 | 100 | 6.7 |
| GNNAK+ [37] | 100 | 100 | 6.7 |
| SSWL+ [36] | 100 | 100 | 6.7 |
| RNM | 100 | 100 | 93.8 |
| I$^2$GNN [15] | 100 | 100 | 100 |
| MAG-GNN | 100 | 100 | 100 |

| Table 2: Cycle counting results. (↓) | | | | |
|---|---|---|---|---|
| Method | 3-CYCLES | 4-CYCLES | 5-CYCLES | 6-CYCLES |
| GIN [32] | 0.3515 | 0.2742 | 0.2088 | 0.1555 |
| RNM | 0.0041 | 0.0129 | 0.0351 | 0.0327 |
| ID-GNN [33] | 0.0006 | 0.0022 | 0.0490 | 0.0495 |
| NGNN [35] | 0.0003 | 0.0013 | 0.0402 | 0.0439 |
| GNNAK+ [37] | 0.0004 | 0.0041 | 0.0133 | 0.0238 |
| I$^2$GNN [15] | 0.0003 | 0.0016 | 0.0028 | 0.0082 |
| MAG-GNN | 0.0029 | 0.0037 | 0.0097 | 0.0286 |

can the expressivity generalize to node-level tasks? **Q3**: How is the RL method's performance on real-world datasets? **Q4**: Does MAG-GNN have the claimed computational advantages?

For the experiment, we update all node tuples simultaneously as it allows more parallelizable computation. More experiment and dataset details can be found in Appendix F.

## 6.1 Discriminative power

**Dataset.** To answer **Q1**, we use synthetic datasets (Accuracy) to test the expressivity of models. (1) EXP contains 600 pairs of non-isomorphic graphs that cannot be distinguished by 1-WL/2-WL bounded GNN. The task is to differentiate all pairs. (2) SR25 contains 15 strongly regular graphs with the same configuration and cannot be distinguished by 3-WL bounded GNN. (3) CSL contains 150 circular skip link graphs in 10 isomorphism classes. (4) BREC contains 400 pairs of synthetic graphs to test the fine-grained expressivity of GNNs (Appendix E). We use the ORD training scheme.

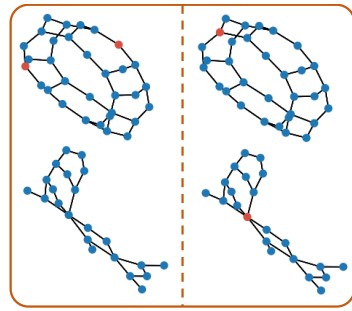

Figure 4: Initial Random Node Marking (Left). MAG-GNN generated markings. (Right)

**Results.** We compare to MPNN [32], Random Node Marked (RNM) GNN with the same hyperparameter search space as MAG-GNN, subgraph GNNs[35, 15, 36, 37], and Non-equivariant GNNs [1] as baselines. Table 1 shows that MAG-GNN achieved a perfect score on all datasets, which verifies our observation and theory. Note that subgraph GNNs like NGNN take at least $|V|$ MPNN runs, while MAG-GNN takes constant times of MPNN runs. However, MAG-GNN successfully distinguished all strongly regular graphs in SR25, which NGNN failed. RNI, despite being universal, is challenging to train and can only make random guesses on SR25. Compared to the RNM approach, MAG-GNN finds more representative subgraphs for the SR25 dataset and performs better. Figure 4 shows a graph in the EXP dataset. We observe that MAG-GNN moves the initial node marking on the same component to different components, allowing better structure learning. Another example of strongly regular graphs is in Appendix E. In Figure

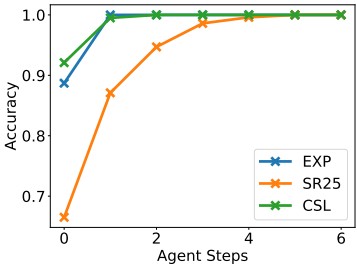

Figure 5: Performance versus the number of RL steps.

5, we also plot the performance of MAG-GNN against the number of MAG-GNN search steps when we only use one node tuple of size two. We see that node tuples from MAG-GNN are significantly more expressive than random node tuples (step 0). On EXP and CSL datasets, MAG-GNN can achieve perfect scores in one or two steps, whereas in SR25, it takes six steps but with a consistent performance increase over steps. We plot the reward curve during training in Appendix E.

## 6.2 Node-level tasks.

**Datasets.** To answer **Q2**, we adopt the synthetic cycle counting dataset, CYCLE (MAE), in Zhao *et al.* [37]. The task is to count the number of (3,4,5,6)-cycles on each node. MPNN cannot count these structures. Node-rooted GNN can count (3,4)-cycles, while only models with expressivity beyond 3-WL can count all cycles. We use the ORD training scheme.

Table 5: QM9 experimental results on all targets. ($\downarrow$)

| Target | RNM | 1-2-3-GNN [23] | PPGN [20] | NGNN [35] | I$^2$-GNN [15] | MAG-GNN |
|--------|-----|----------------|-----------|-----------|----------------|---------|
| Comp. | $O(kT\|V\|^2\|)$ | $O(T\|V\|^4\|)$ | $O(T\|V\|^3\|)$ | $O(T\|V\|^3\|)$ | $O(T\|V\|^4\|)$ | $O(mtT\|V\|^2\|)$ |
| $\mu$ | 0.426 | 0.476 | **0.231** | 0.428 | 0.428 | 0.353 |
| $\alpha$ | 0.306 | 0.27 | 0.382 | 0.230 | 0.230 | **0.226** |
| $\epsilon_{\text{HOMO}}$ | 0.00258 | 0.00337 | 0.00276 | 0.00265 | 0.00261 | **0.00257** |
| $\epsilon_{\text{LUMO}}$ | 0.00269 | 0.00351 | 0.00287 | 0.00297 | 0.00267 | **0.00252** |
| $\Delta\epsilon$ | 0.0047 | 0.0048 | 0.00406 | 0.0038 | 0.0038 | **0.0035** |
| $\langle R^2 \rangle$ | 20.9 | 22.9 | 16.07 | 20.5 | 18.64 | **15.44** |
| $ZPVE$ | 0.0002 | 0.00019 | 0.0064 | 0.0002 | **0.00014** | 0.0002 |
| $U_0$ | 0.281 | **0.0427** | 0.234 | 0.295 | 0.211 | 0.111 |
| $U$ | 0.193 | 0.111 | 0.234 | 0.361 | 0.206 | **0.105** |
| $H$ | 0.384 | **0.0419** | 0.229 | 0.305 | 0.269 | 0.089 |
| $G$ | 0.250 | **0.0469** | 0.238 | 0.489 | 0.261 | 0.116 |
| $C_v$ | 0.177 | 0.0944 | 0.184 | 0.174 | **0.073** | 0.093 |

**Results.** Following Huang *et al.* [15], we say a model has the required counting power if the error is below $0.01$ and report the result in Table 2. We compare to MPNN baseline [32], RNM GNN, node-level subgraph GNNs [33, 37, 35], and edge-level subgraph GNNs [15]. We can see that MAG-GNN successfully counts (3,4,5)-cycles, which indicates that node-marking also helps non-marked nodes to count cycles. We also note that MAG-GNN does not count 6-cycles well, although we use $> 2$-order node tuples. We suspect this is because MAG-GNN takes the average score improvement of nodes as the reward, which might not be the best reward for a node-level task. Even so, we can still observe MAG-GNN's performance improvement over NGNN, which shows that MAG-GNN with larger node tuples indeed increases expressivity. We leave the design of a more proper node-level reward to future works.

## 6.3 Real-world datasets

**Datasets.** To answer **Q3**, we adopt the following real-world molecular property prediction datasets: (1) QM9 (MAE) contains 130k molecules for twelve graph regression targets. (2) ZINC and ZINC-FULL (MAE) include 12k and 250k chemical compounds for graph regression. (3) OGBG-MOLHIV (AUROC) contains 41k molecules, and the goal is to classify whether a molecule inhibits HIV. We use the SIMUL training scheme for a fair comparison to other methods.

Table 3: Transfer learning.

| | ZINC$\downarrow$ | ZINC-FULL$\downarrow$ | MOLHIV$\uparrow$ |
|--------|------|-----------|--------|
| No PRE | 0.106 | 0.030 | 77.12 |
| 3-CYCLES | 0.104 | 0.030 | 76.98 |
| 6-CYCLES | 0.096 | 0.023 | 78.30 |
| SR25 | 0.103 | 0.028 | 77.26 |

Table 4: Inference time.

| Time (ms) | ZINC-FULL | CYCLE | QM |
|-----------|-----------|-------|-----|
| MPNN | 100.1 | 58.4 | 222.9 |
| NGNN | 402.9 | 211.7 | 776.8 |
| I$^2$GNN | 1864.1 | 1170.4 | 3524.0 |
| PPGN | 2097.3 | 1196.8 | 4108.7 |
| MAG-GNN | 385.8 | 155.1 | 704.9 |

**Results.** On QM9 targets, MAG-GNN significantly outperforms NGNN on all targets with an average of $33\%$ MAE reduction. It also outperforms I$^2$-GNN, with partially $>3$-WL expressivity, on most of the targets ($16\%$ average MAE reduction), showing that with much fewer MPNN runs, MAG-GNN can still achieve better performance. This is because despite using fewer subgraphs, we use node tuples with a size greater than two, which grants MAG-GNN the power to distinguish graphs that require higher expressivity. MAG-GNN also performs comparably to 1-2-3-GNN, which simulates 3-WL. We observe that 1-2-3-GNN performs well on the last five targets (which are global properties hard for subgraph GNNs) while outcompeted by MAG-GNN on the rest of the targets. We suspect that 1-2-3-GNN has a constant high expressivity, which might easily lead to overfitting during training. At the same time, MAG-GNN can automatically adjust the expressivity by subgraph selection, reducing the risk of overfitting.

The results on other molecular datasets are shown in Table 6. We see that MAG-GNN outperforms base GNN by a large margin showing its better expressivity. Another important comparison is between MAG-GNN and other non-equivariant GNNs, including PF-GNN, k-OSAN, and RNM. We see that MAG-GNN achieves significantly better results on ZINC, where only models with high expressivity get good performance. This verifies that using an RL agent to capture the inter-subgraph relation is essential in finding expressive subgraphs. MAG-GNN does not perform as well on the

Table 6: Molecular datasets results.($\downarrow$)

| | # Params | ZINC ($\downarrow$) | ZINC-FULL ($\downarrow$) | OGBG-MOLHIV ($\uparrow$) |
|---|---|---|---|---|
| GIN | - | 0.163±0.004 | 0.088±0.002 | 77.07±1.49 |
| PNA | - | 0.188±0.004 | - | 79.05±1.32 |
| k-OSAN | - | 0.155±0.004 | - | - |
| PF-GNN | - | 0.122±0.010 | - | 80.15±0.68 |
| RNM | 453k | 0.128±0.027 | 0.062±0.004 | 76.79±0.94 |
| GSN | - | 0.115±0.012 | - | 78.80±0.82 |
| CIN | ~100k | 0.079±0.006 | **0.022**±0.002 | **80.94**±0.57 |
| NGNN | ~500k | 0.111±0.003 | 0.029±0.001 | 78.34±1.86 |
| GNNAK+ | ~500k | 0.080±0.001 | - | 79.61±1.19 |
| SUN | 526k | 0.083±0.003 | - | 80.03±0.55 |
| KPGNN | 489k | 0.093±0.007 | - | - |
| I$^2$GNN | - | 0.083±0.001 | 0.023±0.001 | 78.68±0.93 |
| SSWL+ | 387k | **0.070**±0.005 | **0.022**±0.002 | 79.58±0.35 |
| MAG-GNN | 496k | 0.106±0.014 | 0.030±0.002 | 77.12±1.13 |
| MAG-GNN-PRE | 496k | 0.096±0.009 | 0.023±0.002 | 78.30±1.08 |

OGBG-MOLHIV dataset. We observe that 1-WL bounded methods PNA also achieves good results on the datasets, meaning that the critical factor determining the performance on this dataset is likely the implementation of the base GNN but not the expressivity. MAG-GNN is highly adaptable to any base GNN, potentially improving MAG-GNN's performance further. We leave this to future work.

We use the PRE training scheme to conduct transfer learning tasks on ZINC, ZINC-FULL, and OGBG-MOLHIV datasets. We pre-train the expressivity datasets shown on the left column of Table 3 and train the attached MPNN head using the datasets on the top row. We see that pretraining consistently brings performance improvement to all datasets. Models pre-trained on CYCLE are generally better than the one pretrained on SR25, possibly due to the abundance of cycles in molecular graphs.

## 6.4 Runtime comparison

We conducted runtime analysis on previously mentioned datasets. Since it is difficult to match the number of parameters for all models strictly, we fixed the number of GNN layers to five and the embedding dimension to 100. We set a 1 GB memory budget for all models and measured their inference time on the test datasets. We use $m = 2$ and $T = 2$ for MAG-GNN. The results in Table 4 show that MAG-GNN is more efficient than all subgraph GNNs and is significantly faster than the edge-rooted subgraph GNN, I$^2$GNN. NGNN achieves comparable efficiency to MAG-GNN because it takes a fixed-hop subgraph around nodes, reducing the subgraph size. However, MAG-GNN outperforms NGNN on most targets with its better expressivity.

**Limitations.** Despite the training schemes in Section 4.1, MAG-GNN is harder to train. Also, MAG-GNN's design for node-level tasks might not be proper. This motivates research on extending MAG-GNN to node-level or even edge-level tasks. We discuss this further in Appendix C.4.

## 7 Conclusions

In this work, we closely examine one popular GNN paradigm, subgraph GNN, and discover that a small subset of subgraphs is sufficient for obtaining high expressivity. We then design MAG-GNN, which uses RL to locate such a subset, and propose different schemes to train the RL agent effectively. Experimental results show that MAG-GNN achieved very competitive performance to subgraph GNNs with significantly less inference time. This opens up new pathways to design efficient GNNs.

**Acknowledgement.** Lecheng Kong, Jiarui Feng, Hao Liu, and Yixin Chen are supported by NSF grant CBE-2225809. Muhan Zhang is partially supported by the National Natural Science Foundation of China (62276003) and Alibaba Innovative Research Program.

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

# Appendix

## A  Proof of theorems

### A.1  Proof of Theorem 1

To prove theorem 1, we leverage the following lemma from [35], and we restate it here.

**Lemma 1.** *For two graphs $G_1$ and $G_2$ that are uniformly independently sampled from all n-node r-regular graphs, where $3 \leq r < \sqrt{2 \log n}$, we pick any two nodes, each from one graph, denoted by $v_1$ and $v_2$ respectively, and do $\lceil (\frac{1}{2} + \epsilon) \frac{\log n}{\log(r-1-\epsilon)} \rceil$-height rooted subgraph extraction around $v_1$ and $v_2$. With at most $\epsilon \lceil \frac{\log n}{\log(r-1-\epsilon)} \rceil$ many layers, a proper message passing GNN will generate different representations for the extracted two subgraphs with probability at least $1 - o(n^{-1})$.*

For completeness, we recall Theorem 1:

**Theorem 1.** *Let $G_1$ and $G_2$ be two graphs uniformly sampled from all $n$-node, $r$-regular graphs where $3 \leq r < \sqrt{2 \log n}$. Given an injective pooling function $R^{(G)}$ and an MPNN $g_p$ of 1-WL expressivity, with a probability of at least $1 - o(1)$, there exists a node (tuple) $v \in V(G)$ whose corresponding node marked graph's embedding, $f_r(G_1, v)$, is different from any node marked graph's embedding in $G_2$.*

*Proof.* Since the pooling function $R^{(G)}$ is injective and the GNN $g$ has 1-WL expressiveness, which is the maximal expressiveness for message-passing GNNs, $f_r$ satisfies the proper GNN requirement in Lemma 1. We also let $g$ have $\epsilon \lceil \frac{\log n}{\log(r-1-\epsilon)} \rceil + d$ layers, where $d = \lceil (\frac{1}{2} + \epsilon) \frac{\log n}{\log(r-1-\epsilon)} \rceil$. Let $v = (v_1)$ on $G_1$, since $v_1$ is marked, we can have the first $d$ layers of $g$ perform breadth-first-search on the graph rooted from $v_1$, and afterward, each node's representation embeds whether it's within $d$ distance from $v_1$. We then let the last $\epsilon \lceil \frac{\log n}{\log(r-1-\epsilon)} \rceil$ layers of $g$ only to perform message passing on the nodes that are within $d$ distance from $v_1$. This can be easily done by implementing an injective COMBINE function. We do the same operation for $v' = (v_2)$ on $G_2$. According to lemma 1, since $f_r$ is proper, the probability that $f_r(G_1, v) \neq f_r(G_1, v)$ is at least $1 - o(n^{-1})$. Then, by union bound, the probability that $f_r(G_1, v) \notin \{f_r(G_2, v') | \forall v' \in V(G_2)\}$ is at least $1 - o(1)$  □

### A.2  Proof of Theorem 2

We first introduce two non-isomorphic graphs in Figure 6.

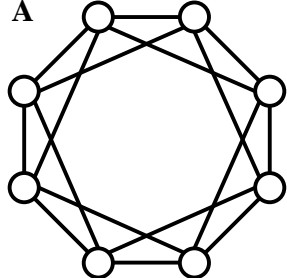 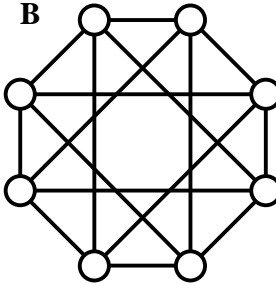

Figure 6: Two CSL graphs.

They are CSL graphs both with 8 nodes, one with a skip factor of 2 and the other with a skip factor of 3. They have the following properties,

- MPNN cannot differentiate them since they are regular graphs.
- They are vertex-transitive graphs. Meaning that all node-rooted subgraphs from one graph are the same.

- The rooted subgraphs from the two graphs are different and can be differentiated by an MPNN.

The properties imply that we cannot use an MPNN to differentiate the two graphs, but we can apply MPNN once on one subgraph from each graph to differentiate them since all subgraphs for one graph are the same.

We name the two CSL graphs A and B and use them to build a super graph. Let $d = 2n$ be a super graph's diameter where $n \in \mathbb{Z}^+$. We randomly permute $n$ type A graphs and $n$ type B graphs to form a graph sequence $S$. An example is $S_e = (A, B, B, A, A, B)$.

We then define $S'$ as the inverse of $S$, that is

$$[S']_i = \begin{cases} A & \text{if } [S]_i = B \\ B & \text{if } [S]_i = A \end{cases} \tag{11}$$

The inverse of $S_e$, $S'_e = (B, A, A, B, B, A)$.

We can then use concatenation to form two new sequences.

$$S^+ = S \oplus S \quad S^- = S \oplus S' \tag{12}$$

For example, $S_e^+ = (A, B, B, A, A, B, A, B, B, A, A, B)$ and $S_e^- = (A, B, B, A, A, B, B, A, A, B, B, A)$. We then use full connections to connect the graphs in a sequence and create a super graph $G(S)$, specifically,

$$G(S) = \{V(G), E(G)\}, \quad V(G) = \cup_i V([S]_i), \quad E(G) = \cup_i (E([S]_i) \cup W(i)))$$
$$W(i) = V([S]_i) \times V([S]_{i+1 \bmod 4n}) \tag{13}$$

The nodes in graph $S[i]$ connect to every node in graph $S[j]$ if the two graphs are adjacent in the sequence or are the head and tail of the sequence. Figure 7 shows the example super graphs $G(S_e^+)$ and $G(S_e^-)$

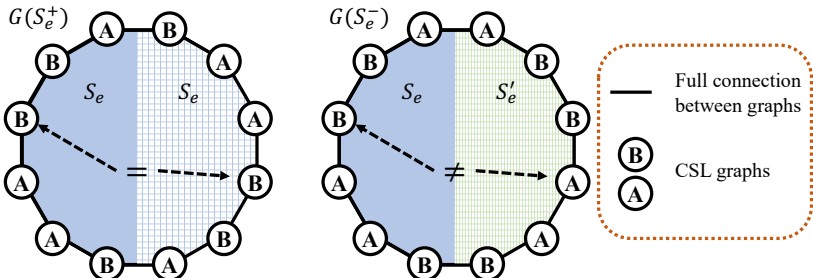

Figure 7: Two example super graphs.

The properties of $G(S)$ is,

- The diameter of the super graph is $2n$.
- $G(S)$ has an equal number of graphs A and B.
- Because adjacent graphs in the super graph are fully connected, adding node marking to graph $[S]_i$ in the super graph does not help us identify any other graphs $[S]_j$ without node marking. To identify $[S]_j$, we must add node marking to $[S]_j$.
- For positive super graphs, $[S]_i = [S]_{i+2n}$, that is, for any graph in the super graph, the graph on its direct opposite shares its type. Whereas for negative super graphs, $[S]_i \neq [S]_{i+2n}$.

Let the task be to discriminate between positive super graphs and negative super graphs. We then prove that MAG-GNN is superior in this case to show strictness.

*Proof.* We consider the case where $k = 2$ and $m = 1$. That is, we use one node tuple of size two. Let MAG-GNN implement the following policy: *move one node in the node tuple to the node in the graph that is farthest away from the other node.* This is easy to be implemented by MAG-GNN since

both nodes are marked, and we can compute the distance from all nodes in the graph to the two nodes with a sufficient number of MPNN layers. According to the properties of $G(S)$, MAG-GNN will move the node tuples so that they sit on the opposite graphs in the super graph. They identify the type of the two graphs, determining if $G(S)$ is positive. Note that, in this case, we execute MAG-GNN **once** to obtain the discriminating node tuple independent of the super graph diameter.

On the other hand, if we use random actions, because node markings only help identify the graphs that the markings land on, marking nodes on any location other than the two opposite graphs will not help identify the super graph. The probability of random actions choosing the opposite graphs is $\frac{1}{4n-1}$. Consequently, the expected number of random actions required to identify the super graph is $4n - 1$. $\qquad\square$

### A.3 Proof of Theorem 3

We first describe the PF-GNN algorithm; we refer the readers to Dupty *et al.* [7] for more details.

PF-GNN starts with a set of particles for a graph $G$,

$$\langle (G, H_1^m), w_1^k \rangle_{m=1:M} \tag{14}$$

where $H$ is the node embedding of $G$ using a MPNN $g$. For particle $m$ in step $t$, PF-GNN uses a policy function $P(V|H_t^m)$ to compute a distribution of nodes and sample a node to individualize. After using an MLP to update the embedding of the individualized node, PF-GNN uses GNN to refine all node embeddings. Specifically,

$$v \sim P(V|H_t^m), [H_t^m]_v = \text{MLP}([H_t^m]_v), H_{t+1}^m = g(H_t^m) \tag{15}$$

Then PF-GNN updates the particle weights as follows,

$$w_{t+1}^m = \frac{f_{obs}(H_{t+1}^m) \cdot w_t^m}{\sum_k f_{obs}(H_{t+1}^m) \cdot w_t^m} \tag{16}$$

where $f_{obs}$ is a function that measures the quality of the refined embeddings to a scalar value. PF-GNN then performs particle resampling from the distribution of $\langle w_{t+1}^m \rangle_{m=1:M}$ and reset weights to $1/M$. Eventually, all $H_{t+1}^m$ node representations undergo a readout function to generate graph embedding.

*Proof.* To prove MAG-GNN captures PF-GNN, we need to show that in each step, the node representation generated by PF-GNN and MAG-GNN are identical, resulting in the same node sampling distribution, resampling process, and weight update. Specifically, for a $M$-particle, $T$-step PF-GNN, we implemented a MAG-GNN with $M$ node tuples of order $T$ using $T$ RL steps. The node tuples are still randomly initialized. The state matrix $W$ is $k$-by-$(1+T)$, where $[W]_{m,i} = 0, i \neq T+1$ indicates the $i$-th node in node tuple $m$ has not been individualized, $[W]_{m,i} = j, i \neq T+1$ indicates the $i$-th node in node tuple $m$ is individualized at time step $j$, and $[W]_{m,T+1}$ is the weight in PF-GNN.

Let $\boldsymbol{v} = (v_1, ..., v_t)$ be the sequence of nodes that PF-GNN used to individualize. We first note that there exists an MPNN applied onto the node marked graph $G(\boldsymbol{v})$ with a sufficient number of layers that output the same value as the final output, $H_t$, of PF-GNN. Specifically, let $T$ be the number of layers of $g$ in the MPNN, and then we can build a $tT + t$-layers MPNN $g_p$ such that it alternates between $T$ layers MPNN that is a copy of $g$ and one layer MPNN that only update node features on marked nodes using the MLP of PF-GNN. Then, as long as we know the sequence, we can mimic a PF-GNN with an MPNN on node-marked graphs.

We then implement an update function $f_W$ to log the step when a node $v_i$ in the tuple $m$ is marked that is $[W]_{m,t} = t$. We then set the Q-network to never output positive scores to the tuple index whose corresponding entries in $W$ are greater than zero so that we fix the individualized nodes. $f_W$ also update $[W]_{m,T+1}$ to be equal to the weight after the resampling process. Then, by using the $tT + t$-layers MPNN consists of $g$ as $g_p$ and another $tT + t$-layers MPNN consists of the MPNN of $P(V|H_t)$ as $g_{rl}$, MAG-GNN can replicate the output of PF-GNN. $\qquad\square$

## B Introduction to deep Q-learning

Deep Q-learning (DQN) is a neural network variant of Q-learning which is a popular algorithm for solving the Markov decision process (MDP). MDP is defined by a tuple $(S, A, P, R, \gamma)$, where:

- $S$ is a set of states.
- $A$ is a set of actions.
- $P$ is a transition function that defines the probability of outcomes states after a particular action. For MAG-GNN, the transition is deterministic.
- $R$ is a reward function defining the reward received by an action in a particular state.
- $\gamma$ is a discount factor determining future rewards' importance.

Agents in an MDP aim to learn a policy that maps states to actions with maximal expected rewards, and Q-learning generates such agents. Specifically, Q-learning maintains a Q-table of Q-values $Q(s,a)$, representing the expected cumulative rewards of taking action $a$ in state $s$. During training the Q-values are updated by,

$$Q(s,a) = Q(s,a) + \alpha(r + \gamma * \max_{a'}(Q(s',a')) - Q(s,a)) \tag{17}$$

where $\alpha$ is the learning rate, $r$ is the instant reward of action $a$ in $s$ and $s'$ is the resulting state of $a$. In more complex systems, keeping enormous Q-tables for all actions and states is infeasible, and DQN uses deep neural networks to build a Q-network to approximate $Q(s,a)$. The Q-network takes $s$ as input and output $Q(s,a)$ for all possible actions $a$. A loss function to train the Q-network is,

$$\mathcal{L} = \mathbb{E}[(r + \gamma * \max_{a'} Q(s',a') - Q(s,a))^2] \tag{18}$$

A widely adopted method to train the Q-network is the Experience Replay. In each step, the RL agent interacts with the environment to store experience tuples $E = (s,a,r,s')$, and sample a set of previous experiences to train the Q-network. This technique effectively improves the sample efficiency. We adopt $\epsilon$-greedy training strategy, where the agent takes random actions with a probability of $\epsilon$. We start with $\epsilon = 1$ and gradually decrease it to zero. This ensures that the agent explore the graph and learns the good policy.

## C  More discussion on the design of MAG-GNN

### C.1  Action space

In Section C.1, we propose an action to substitute one node in the node tuple with another node. Such action requires a $O(m|V|k)$ Q-table for selecting the node tuple ($m$), selecting the node tuple index ($k$), and selecting the node in the graph. We also propose not to select node tuple to reduce the $m$ factor, then the action in 7 becomes

$$\boldsymbol{v}_i' = a_{j,l}(\boldsymbol{v}_i) = ([\boldsymbol{v}_i]_1, ..., [\boldsymbol{v}_i]_{j-1}, v_l, [\boldsymbol{v}_i]_{j+1}, ...[\boldsymbol{v}_i]_k) \tag{19}$$

The action space becomes $A = [k] \times \mathcal{V}$. In such a case, we need to enforce an order to update each node tuple sequentially or simultaneously update them. In the experiments, we choose to update all node tuples for better parallelizability simultaneously.

A seemingly more natural design choice is directly replacing the current node tuple with another one. However, it has $O(|V^k|)$ possible actions for one node tuple. Its corresponding Q-table size is also exponential to $k$. The cost of computing such a table may equal that of running the complete subgraph GNN, which violates our motivation for complexity reduction.

The state matrix $W$ serves as a tracker of previous states. The choice of update function $f_W$ can vary. It can simply be a mean pooling function on the embeddings of marked nodes. It can also be an LSTM pooling function. While using matrix $W$ has good theoretical properties as demonstrated in Appendix A, we found it is not as helpful practically and becomes a confounding factor under the SIMUL training scheme because the states matrix stored in the memory does not properly reflect the prediction GNN. Hence, we use zero-initialized $W$ and an identity update function during training.

### C.2  Training paradigms

We provide pseudo-codes for each training paradigm of MAG-GNN. All training paradigms are based on experience replay, where in each step the agent obtains experiences from the environment, stores the experiences into memory, and retrieves other experiences from memory to optimize the

---

**Algorithm 1** RL-Experience

---

**Input:** Node tuples $U$, batch of data $B$, buffer $D$, RL agent $a$ and GNN $g_{rl}$
**Output:** New node tuples.
1: Use $a$ and $g_{rl}$ to generate a new node tuples $U'$ according to equation (9) and (10)
2: Obtain rewards $r$ from $U'$, $U$, and $B$ according to equation (8)
3: Store the experience tuple $(U', U, B, r)$ to $D$
4: **return** $U'$

---

---

**Algorithm 2** ORD-Train

---

**Input:** A target dataset of $(\mathcal{G}, \mathcal{Y})$. A graph prediction model $f_p$, an RL agent $a$ with RL GNN $g_{rl}$.
**Output:** A trained RL agent $a$ that output the most discriminative node tuples and a corresponding prediction GNN $f_p$.
 1: **while** $f_p$ is not converged **do**
 2:      $B \leftarrow$ a batch from $(\mathcal{G}, \mathcal{Y})$
 3:      Sample random node tuples $U$ for graphs in $B$
 4:      Optimize $f_p$ according to equation (4) w.r.t $B$ and $V$.
 5: **end while**
 6: $D \leftarrow$ memory buffer
 7: Call RL-Experience until $D$ has enough memory
 8: **while** $g_{rl}$ is not converged **do**
 9:      $B \leftarrow$ a batch from $(\mathcal{G}, \mathcal{Y})$
10:      Sample random node tuples $U$ for graphs in $B$
11:      RL-Experience($U$, $B$, $D$, $a$, $g_{rl}$)
12:      Sample experience tuples $E$ from $D$
13:      Optimize $g_{rl}$ using the $E$
14: **end while**
15: **return** $f_p$, $a$, $g_{rl}$

---

Q-learning agent. Since we are training the agent (RL GNN) while the environment (prediction GNN) is evolving, we propose different training paradigms to better model both sides. Different training paradigms differ mainly in the schedule to train the two sides. ORD paradigm first trains the environment without the agent and then trains the agent in the fixed environment. SIMUL paradigm trains the agent and environment together, where an environment training step is followed by the agent collecting experience from the current environment and optimizing according to past experience. The SIMUL paradigm is proposed because the trained environment might overfit the data and hence give incorrect rewards to the agent. We provide pseudo-codes for the paradigms.

### C.3    Other sampling-based methods

The advantage of MAG-GNN over other subgraph sampling methods, including PF-GNN [7] and DropGNN [24], is that it effectively chooses the most discriminative subgraphs. DropGNN drops nodes with equal probability; PF-GNN uses an MPNN to model the drop probability of nodes which reduces to the uniform distribution when the graph is regular and has no features. According to Figure 2, there is still a high chance that the randomly sampled subgraphs are not informative. On the contrary, MAG-GNN starts with random subgraphs but uses previous knowledge about the correlation between subgraphs to refine the likelihood of a subgraph being informative. RL training helps the model acquire such knowledge. Just as Theorem 2 points out, previous methods are essentially MAG-GNN with random action. In contrast, learned actions can significantly improve efficiency and effectiveness.

### C.4    Limitations

As mentioned in Section 4.1, MAG-GNN does not train as well on datasets with labels, and the training time is longer than subgraph GNNs despite better inference complexity. Note that we are employing the basic Q-learning techniques in MAG-GNN to demonstrate the framework's power,

**Algorithm 3** SIMUL-Train

**Input:** A target dataset of $(\mathcal{G}, \mathcal{Y})$. A graph prediction model $f_p$, an RL agent $a$ with RL GNN $g_{rl}$.
**Output:** A trained RL agent $a$ that output the most discriminative node tuples and a corresponding prediction GNN $f_p$.
1: $D \leftarrow$ memory buffer
2: Call RL-Experience until $D$ has enough memory
3: **while** $f_p$ and $g_{rl}$ are not converged **do**
4:     $B \leftarrow$ a batch from $(\mathcal{G}, \mathcal{Y})$
5:     Sample random node tuples $U$ for graphs in $B$
6:     $U'$=RL-Experience($U$, $B$, $D$, $a$, $g_{rl}$)
7:     Optimize $f_p$ according to equation (4) w.r.t $B$ and $U'$
8:     $B' \leftarrow$ a batch from $(\mathcal{G}, \mathcal{Y})$
9:     Sample random node tuples $U$ for graphs in $B'$
10:     RL-Experience($U$, $B$, $D$, $a$, $g_{rl}$)
11:     Sample experience tuples $E$ from $D$
12:     Optimize $g_{rl}$ using the $E$
13: **end while**
14: **return** $f_p$, $a$, $g_{rl}$

**Algorithm 4** PRE-Train

**Input:** A target dataset of $(\mathcal{G}, \mathcal{Y})$. A graph prediction model $f_p$, an RL agent $a$ with RL GNN $g_{rl}$ trained with either SIMUL or ORD on a different dataset.
**Output:** A trained RL agent $a$ that output the most discriminative node tuples and a corresponding prediction GNN $f_p$.
1: **while** $f_p$ is not converged **do**
2:     $B \leftarrow$ a batch from $(\mathcal{G}, \mathcal{Y})$
3:     Sample random node tuples $U$ for graphs in $B$
4:     $U'$=RL-Experience($U$, $B$, $D$, $a$, $g_{rl}$)
5:     Optimize $f_p$ according to equation (4) w.r.t $B$ and $U'$
6: **end while**
7: **return** $f_p$, $a$, $g_{rl}$

and we can adopt more robust training methods from the large RL toolbox to improve training time. We leave this to future work.

An advantage of subgraph GNN is its equivariance property. Given the same graph up to isomorphism, subgraph GNNs always generate the same embedding for the graph, whereas MAG-GNN sacrifices such property for better efficiency. However, MAG-GNN still provides a higher level of certainty than other sampling-based and randomization-based non-equivariant GNNs, because when the initial node marking is fixed, MAG-GNN can also generate the same embedding for the graph. Then, suppose we can deterministically generate initial node markings; we also grant MAG-GNN equivariance. Such initial node marking can be computed from canonical labeling or previous knowledge about the graph.

While we see MAG-GNN's potential in node-level tasks in Section 6, MAG-GNN is primarily defined at the graph level, and its reward and actions might not be the most appropriate for node-level tasks. We conduct the evaluation in Section 3 on the Cycle counting dataset; we observe that even for 6-CYCLE counting, there exist node marked graphs that result in near zero error while our Q-learning framework fails to identify such node tuple. We suspect this is because the MAG-GNN's reward is defined at graph-level, and taking a plain average of node error does not faithfully capture the environment. This leads to exciting research paths extending MAG-GNN to node-level or even edge-level tasks.

Table 7: BREC dataset results (↑)

| Model | Basic (60) | | Regular (140) | |
| | Number | Accuracy | Number | Accuracy |
| --- | --- | --- | --- | --- |
| NGNN [35] | 59 | 98.3 | 48 | 34.3 |
| SSWL+ [36] | 60 | 100 | 50 | 35.7 |
| I$^2$GNN [15] | 60 | 100 | 100 | 71.4 |
| OSAN [26] | 56 | 93.3 | 8 | 5.7 |
| MAG-GNN | 49 | 81.6 | 61 | 43.5 |

## D Complexity analysis

The complexity of MAG-GNN is straightforward. It runs MPNN multiple times; hence, the complexity is a multiple of MPNN ($O(T|V^2|)$). Specifically, for a MAG-GNN with $m$ $k$-tuples that runs $t$ steps, there is a total number of $O(mt)$ MPNN runs, incurring $O(mtT|V^2|)$ total complexity. The complexity of computing the Q-table is equal to the size of the Q-table, which is $O(m|V|k)$. We compute Q-tables $O(t)$ times, and the overall complexity for computing the Q-table is $O(mkt|V|)$. The overall complexity is $O(mkt|V| + mtT|V^2|)$. In practice, $O(mkt) < O(T|V|)$ as can be observed in Table 10 and hence the complexity can be reduced to $O(mtT|V^2|)$. Note that subgraph GNNs have the complexity of $O(T|V^k|)$, which grows exponentially with $k$, while MAG-GNN's complexity grows linearly with $k$ even when $k$ is large.

In equation 9, we propose to use graph embedding function $f_p$ to represent the current state; however, an alternative is to use $f_{rl}$ to describe the current state. This allows us not to compute graph embedding for prediction but only compute that for Q-table. This does not change the complexity but can reduce the number of MPNN runs in half, improving efficiency.

## E More experimental results

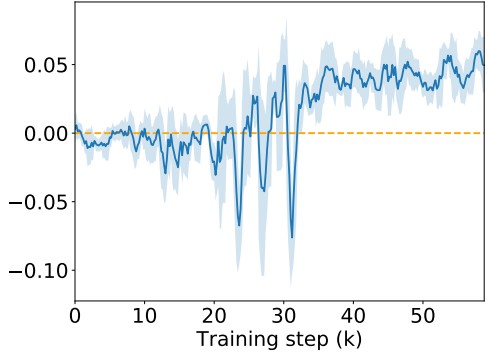

Figure 8: Reward increases over time.

Table 7 shows the results on the BREC dataset's basic and regular graph subsets [28]. BREC dataset contains 400 pairs of graphs that require different levels of expressiveness to distinguish. The regular graph dataset contains strongly regular and distance-regular graphs that cannot be distinguished by 3-WL GNNs. We apply a fixed 2-node-tuple MAG-GNN to these pairs and test 1000 times with different initial node tuples. We say the model distinguishes the pair if the model correctly classifies the pair at least 950 times. We then count the ratio of correctly distinguished pairs. We observe that MAG-GNN falls short on the basic graph, possibly due to the excessive expressiveness and the difficulty it poses to training. However, we also observe that MAG-GNN performed well on regular graphs, outperforming NGNN and SSWL+ with <3-WL expressiveness. While outcompeted by I$^2$GNN, MAG-GNN is considerably more efficient. We also noticed that MAG-GNN significantly improves the performance on regular graphs over OSAN, further validating that RL-selected subgraphs carry more information.

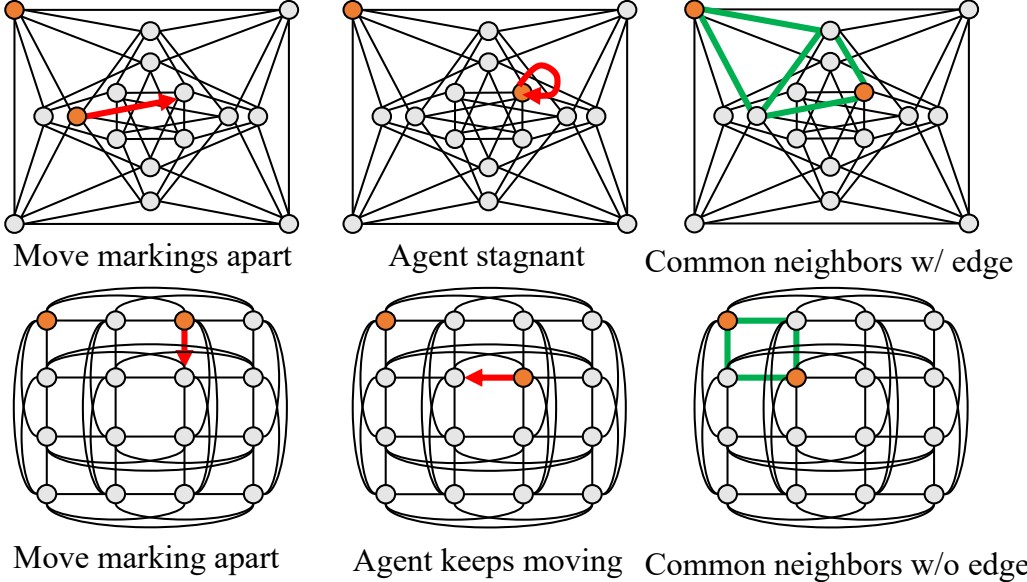

| Move markings apart | Agent stagnant | Common neighbors w/ edge |
| Move marking apart | Agent keeps moving | Common neighbors w/o edge |

Figure 9: MAG-GNN example on Shrikhande graph (Top) and Rood 4-by-4 graph (Bottom). Red arrows show how MAG-GNN moves the node tuples. Green edges show the difference in the subgraphs defined by the resulting node tuples.

In Figure 8, we show the reward improvement over the training step on the SR25 dataset. We plot the mean reward of every 500 steps and plot the standard deviation of the 500 steps by the shaded range. We can see that the reward is close to zero at the initial stage of training because the $\epsilon$ is set to 1, and the agent essentially takes random actions. After a period of searching where the reward stays at around 0, the output node tuples gradually become informative. Eventually, the step-average reward converges at 0.05. We see a few sudden drops in reward during training due to the "overestimation bias" commonly seen in RL. Nevertheless, we see that the agent can correct the mistake and eventually converge. More advanced deep Q-learning techniques like double Q-learning might resolve the issue, and we leave this as future work.

We conducted the experiments on the Shrikhande graph versus the Rook 4-by-4 graph toy example. The task is to differentiate the two strongly regular graphs with the same configuration and hence cannot be distinguished by MPNN and node-rooted subgraph GNNs. We train a MAG-GNN with 2-node markings to break the 3-WL expressivity bound. After training, the MAG-GNN generates the trajectory given the initial 1-distance random node markings as shown in Figure 9. We can observe that MAG-GNN first learns to push the two marked nodes apart in both graphs. However, in the Shrikhande graph, after it finds a marked node pair whose common neighbors are connected, the agent becomes stagnant, while in the Rook 4-by-4 graph, the agent is never stagnant and keeps moving one marked node, because none of the common neighbors of 2-distanced node pairs in the Rook 4-by-4 graph are connected. Such information is easy to be learned by a downstream MPNN.

# F   More experiment details

## F.1   Datasets statistics

We took the CYCLE dataset from [37] repository. All training and testing are on standardized targets following the previous practice [37, 15]. The task is regression, and we use continuous value to approximate the discrete number of cycles. We use the QM9 dataset provided by Pytorch-Geometric [9], and we use a train/valid/test split ratio of 0.8/0.1/0.1. All QM9 targets are standardized during training but mapped back to their actual value for prediction and testing. We use the ZINC dataset provided by Pytorch-Geometric [9] and use the official split. We take OGBG-MOLHIV dataset from the Open Graph Benchmark package [14] and use their official split. For CSL and EXP, we follow

Table 8: Hyperparameters Summary

| Method | QM9 | ZINC | ZINC-FULL | CYCLES |
|---|---|---|---|---|
| Learning rate | | 0.001 | | |
| Weight decay | 0 | 0 | 0 | 0 |
| # GNN Layers | 5 | 6 | 6 | 6 |
| Jumping Knowledge | Sum | Sum | Sum | Sum |
| m | $\{2,3,4\}$ | $\{2,3,4\}$ | $\{2,3,4\}$ | $\{2,3,4\}$ |
| k | $\{3,4,5,6\}$ | $\{3,4,5,6\}$ | $\{3,4,5,6\}$ | $\{3,4,5,6\}$ |
| t | 2 | 2 | 2 | 2 |
| # Epochs | 2000 | 4000 | 500 | 5000 |
| Batch size | 512 | 512 | 512 | 500 |
| Memory size | $3M$ | $300K$ | $300K$ | $300K$ |
| Sync rate | 3000 | 3000 | 3000 | 3000 |
| # runs | 4 | 10 | 3 | 10 |

Table 9: Hyperparameters Summary (cont.)

| Method | MOLHIV | EXP | SR25 | CSL |
|---|---|---|---|---|
| Learning rate | | 0.001 | | |
| Weight decay | 0.001 | 0 | 0 | 0 |
| # GNN Layers | 6 | 4 | 4 | 4 |
| Jumping Knowledge | Sum | None | None | None |
| m | $\{2,3,4\}$ | $\{1,2,3\}$ | $\{1,2,3\}$ | $\{1,2,3\}$ |
| k | $\{3,4,5,6\}$ | $\{1,2,3\}$ | $\{1,2,3\}$ | $\{1,2,3\}$ |
| t | 2 | 4 | 4 | 4 |
| # Epochs | 1000 | 500 | 10000 | 500 |
| Batch size | 128 | 64 | 32 | 64 |
| Memory size | $300K$ | $30K$ | $30K$ | $30K$ |
| Sync rate | 3000 | 100 | 150 | 150 |
| # runs | 10 | 10 | 10 | 10 |

Table 10: Datasets statistics.

| Dataset | #Graphs | Avg. #Nodes | Avg. #Edges | Task type |
|---|---|---|---|---|
| CYCLE | 5,000 | 18.8 | 31.3 | Node regression |
| QM9 | 130,831 | 18.0 | 18.7 | Graph regression |
| ZINC | 12,000 | 23.2 | 24.9 | Graph regression |
| ZINC-FULL | 249,456 | 23.1 | 24.9 | Graph regression |
| ogbg-molhiv | 41,127 | 25.5 | 27.5 | Graph classification |

previous practices and use 10-fold cross-validation to train the model and report the mean results. For SR25, since there are only 15 different graphs, we follow previous practices and use the complete data for training, valid, and test sets to examine the expressivity of the model.

### F.2 Training details

We summarize the hyperparameters used for different datasets in Table 8 and 9.

Curly bracket means the range of hyper-parameters we search. Memory size is the size of the memory buffer for experience and replay. Sync rate is the number of optimization steps before the target Q-network sync with the train Q-network. For ZINC datasets, we additionally use learning rate decay with a rate of 0.5 for every 200 epochs. All models are implemented in DGL [29] and PyTorch [25]. We only use Pytorch Geometric [9] to get the datasets.

Since there is a random factor in MAG-GNN (initial node tuples), all tests are repeated ten times, and we take the average score as the test score of one run. We ran experiments multiple times with different random seeds to take the average scores.

