# OpenReview forum: "MAG-GNN: Reinforcement Learning Boosted Graph Neural Network"
_NeurIPS.cc/2023/Conference — NeurIPS 2023 poster_

### Official Review · Reviewer_9Mgd · 2023-07-04

**Soundness:** 3 good
**Presentation:** 1 poor
**Contribution:** 3 good
**Rating:** 4
**Confidence:** 2

**Summary:**

The authors present MAG-GNN, a subgraph-based Graph Neural Network (GNN) method that utilizes reinforcement learning to effectively identify discriminative subgraphs. They empirically assess the performance of MAG-GNN using both synthetic and real-world datasets.

**Strengths:**

- The main elements of the RL framework are properly explained.
- The empirical section includes clear and valuable research questions.

**Weaknesses:**

1. The writing is significantly lacking in clarity and context.
2. The presentation of results lacks clarity.

**Questions:**

1. Please provide clarification on what "this property" refers to in line 60.


2. Can you confirm if \mathbf{v} represents a set of k nodes in line 88? It seems so, but the notation is not defined.

3. In line 88, it would be helpful to provide context on why a subgraph GNN creates |V^k(G)| copies of the graph and an intuition of what this quantity represents.

4. In Equation (2), could you specify what j represents?

5. Could you explain what N refers to in Equation (3)?

6. In line 136, when you mention a "2-node-marked graph," could you provide a general definition of a node-marked graph?

7. What is meant by a "GNN run" or "MPNN run"?
8. The explanation in the text of Figure 2 is quite complex. Additional information from the authors would be appreciated to improve comprehension.

9. Is \mathcal{V}^k the same as V^k(G) mentioned in line 172? If so, it would be helpful to use standardized notation to help in understanding.

10. In section 4.1 on training MAG-GNN, it is challenging to grasp all the training procedure details solely from the text. It would be beneficial to include an algorithm that illustrates the training procedure.

11. The captions of Table 1 and Table 2 do not indicate which metrics are being presented.
12. What do the yellow cells indicate in Table 2?
13. All the tables and figures in the experimental section lack error bars.
14. How does the runtime of MAG-GNN vary with m and T in section 6.4 on runtime comparison?


**Limitations:**

The authors have acknowledged the limitations of their proposed approach.

---

> ### Author Rebuttal · Authors · 2023-08-07
>
> We are grateful to reviewer 9Mgd’s effort in helping us improve our work and paper. We address the reviewer’s concerns and questions as follows.
>
> - Q1: “The property” refers to many graphs’ properties that a discriminative small subset, rather than the whole set, of a graph’s subgraphs is sufficient in identifying the graph. More detailed explanation in the previous paragraph.
>
> - Q2: Here, $\boldsymbol{v}$ represents a node tuple of k nodes. It’s different from a node set, as it allows repetition and is ordered. For example, when k=2, a node tuple $(\boldsymbol{v}_1, \boldsymbol{v}_1)$ is allowed, and node tuples $(\boldsymbol{v}_1, \boldsymbol{v}_2)$ and $(\boldsymbol{v}_2, \boldsymbol{v}_1)$ are different. We follow the convention of using the power of  $V$ to represent the set of node tuples.
>
> - Q3: The power of **node-rooted subgraph** GNNs over regular GNNs is that they do not directly run MPNN on the graph. Instead, they create $|V|$ copies of the graph and apply MPNN onto each copy, where $|V|$ is the number of nodes. Such an approach requires each copy to be associated with a node in the graph, making each copy unique. A widely-adopted way is to use the subgraph marking technique introduced in lines 90-92. The unique marking to each copy allows the corresponding nodes to learn local structures that plain GNNs cannot. Ultimately, the information of all copied graphs is pooled to form the final representation. Since GNN with $|V|$ graphs can be more expressive than regular GNNs, later work proposes to use **2-node-rooted subgraph** GNNs that create $|V^2|$ graph copies. This approach results in an even stronger GNN than node-rooted subgraph GNN. Naturally, we can keep extending subgraph GNNs to use more subgraphs, which generalize to **k-node-rooted subgraph** GNNs. Theoretical work finds out that subgraph GNNs expressivity increases as k increases. *Because there are $|V^k|$ unique node tuples, we can thus have $|V^k|$ graph copies with unique markings.*
>
> - Q4: We will clarify using an example:
>
> Let G be a graph consisting of nodes $v_1$, $v_2$, $v_3$, $v_4$, and $k=2$, meaning we assign unique labels to 2-node-rooted graphs. Let the node tuple be $(v_2, v_4)$ , and let $c^+=1$ and $c^-=0$. The resulting unique node labeling matrix $X$ is
>
> | COL 1 | COL 2 |
> | --- | --- |
> | 0 | 0 |
> | 1 | 0 |
> | 0 | 0 |
> | 0 | 1 |
>
> Because $v_2$ is the first node ($j=1$) in the node tuple, we have $X_{2,1}=1$, and $v_4$ is the second node ($j=2$) in the node tuple, we have $X_{4,2}=1$. $j$ represents the index in the node tuple.
>
> - Q5: N is not for the natural number set. It represents (**N**)ode-level pooling function to distinguish it from (**G**)raph-level pooling function, as described in line 98.
>
> - Q6: Random 2-node-marked graphs represent the graph copies labeled by random 2-node tuples under the labeling described in equation (2). A 2-node-marked graph creates a copy of the original graph, draws a random 2-node tuple from $V^2$ , and labels the graph according to equation (2). In the paper, we use the uniquely labeled/marked graph as a general form of subgraph because previous work points out that a labeled graph is no less expressive than the subgraph [1].
>
> - Q7: One GNN/MPNN run means using GNN or MPNN to process a graph/subgraph once to get its node/graph embedding. If we run GNN/MPNN on a set of $|V|$ graphs, we say we perform $|V|$ GNN runs on the set of graphs.
>
> - Q8: Figure 2 shows the prediction score of individual 2-node-marked graphs in the SR25 dataset. Recall that a graph $G$ in the SR25 dataset has 25 nodes; hence, there are $25^2=625$ unique 2-node-marked graphs. Following equation (4), we can compute the prediction score of each 2-node-marked graph to the class of $G$, resulting in 625 prediction scores. We sort these scores and plot them in Figure 2. Because SR25 has 15 graphs, Figure 2 has 15 lines. The figure shows that among all possible 2-node-marked graphs of graph $G$, many individually identify $G$ from the rest of the graphs in the dataset.  Such an experiment shows that one of the most challenging datasets has the desirable property that a small set of subgraphs is discriminative enough, which motivates our RL framework to find such subgraphs.
>
> - Q9: They are not the same. $V$ refers to the set of nodes in a specific graph $G$. However, graphs have different node sets. Hence, we use $\mathcal{G}$ to represent the set of all graphs and $\mathcal{V}$ to represent the set of all possible nodes appearing in $\mathcal{G}$. Consequently, $\mathcal{V}^k$ represents all possible k-node tuples of all graphs.
>
> - Q10: We thank the reviewer’s suggestion and include the pseudocode in the pdf attached to the general response.
>
> - Q11: We thank the reviewer for pointing out the unclarity. While we already included the evaluation metric description in the experimental results section, we will include this when more space is permitted in the revision.
>
> - Q12: Yellow cells indicate MAE smaller than 0.01, as mentioned in line 350. We will include this information in the caption in the revision.
>
> - Q13: We omit the error bar following previous works as the variance is minimal [2]. The results for molecular data (Table 6) included in the appendix reports standard deviations. We will move it to the main paper to account for model's randomness.
>
> - Q14: We present the inference time of MAG-GNN on the QM dataset following the setup in section 6.4.
>
> | Time(ms) | m=2 | m=3 | m=4 |
> | --- | --- | --- | --- |
> | t=2 | 704.9 | 847.9 | 959.2 |
> | t=3 | 925.5 | 1275.1 | 1445.5 |
> | t=4 | 1489.0 | 1743.6 | 1875.4 |
>
> We can see that under all these circumstances, MAG-GNN is still faster than I^2GNN and PPGN.
>
> [1] Zhang, Bohang, et al. "Rethinking the expressive power of gnns via graph biconnectivity." arXiv preprint arXiv:2301.09505 (2023).
>
> [2] Huang, Yinan, et al. "Boosting the Cycle Counting Power of Graph Neural Networks with I $^ 2$-GNNs." *arXiv preprint arXiv:2210.13978* (2022).

---

> ### Author Response · Authors · 2023-08-14
>
> Dear Reviewer 9Mgd,
>
> Thank you for acknowledging our contribution,  and we greatly appreciate your time to provide invaluable comments on our work. We just want to check and see if our rebuttal clears the confusion. In the rebuttal, we clarify the notation and terminology we adopted and provide pseudocode for better comprehension of the paper. We also include a runtime table varying $m$ and $k$ to understand the complexity of our method better. Following your suggestion, we plan to improve the presentation of the paper by clearly stating the assumption and providing a more detailed background. We are happy to answer any more questions you have and look forward to your reply.
>
> Submission 3693 authors

---

### Official Review · Reviewer_jrFK · 2023-07-05

**Soundness:** 2 fair
**Presentation:** 3 good
**Contribution:** 2 fair
**Rating:** 6
**Confidence:** 4

**Summary:**

The paper considers the problem of increased complexity of subgraph GNNs with respect to standard MPNNs, due to the enumeration of the entire bag of subgraphs for every given graph.

The authors first study the need of enumerating all subgraphs, and show that there exist graphs where using a small subset of subgraphs have the same discriminative power as the entire bag of subgraphs. Motivated by these cases, the authors propose an RL-based approach, named MAG-GNN, to find the best subset of subgraphs to be employed in the task at hand.

The method starts with a subset of subgraphs that are randomly selected. Then, it employs a DQN framework, where the Q-network is parametrized by a subgraph GNN, to replace each subgraph in the subset with a another subgraph. The new subgraph is obtained by replacing only one index of the node tuple identifying the subgraph. Upon reaching a maximum number of steps, the obtained subgraph set is passed through a downstream subgraph GNN for the final predictions.

**Strengths:**

1. The problem of reducing the complexity of subgraph GNNs without trading off performances is significant. The authors identify graphs where a small subset of subgraphs has the same discriminative power as the full bag of subgraphs, which motivates the search for such subsets.
2. The combination of RL with subgraph GNNs is original, and represent a natural way of learning to select subgraphs.

**Weaknesses:**

1. **The paper misses a thorough comparison with a random policy**, where a random subset of subgraphs is randomly selected at every forward pass. The authors consider tangentially the question in the experimental evaluation (Q1) but fail to answer in details. For every dataset you consider, there should be a table entry reporting the performance of the random baseline. **Such baseline should have the same $k$ and the same $m$ that you consider**. Furthermore, it should be based on the same architecture that MAG-GNN uses as downstream prediction model.  Only with such comparison we can conclude that MAG-GNN performs better than random. Note that:
    - The answer of Q1 is insufficient, both because you consider only one dataset, and also because you focus only on one node tuple ($m=1$). Please include a table entry for every dataset.
   - The experiment in Appendix E reports a comparison with a random baseline that uses ensemble predictions. However, it would make more sense to aggregate the representations of the subgraphs and _then_ perform a prediction, an approach that is indeed followed by subgraph GNNs (as reported in Equation 3). The ensemble method instead predicts using each subgraph separately and then aggregates the predictions, which is clearly less powerful. Please include a comparison with random that does not use the ensemble approach, but that aggregates the representations of the $m$ subgraphs to obtain a graph representation, and then performs the prediction. Note that you should evaluate the same $m$ and $k$ values that you sweep on for MAG-GNN (up to 4 and 6 respectively).

2. **The paper misses comparisons with the I-MLE approach proposed in $k$-OSAN** to select subgraphs, which has exactly the same goal.

3. **The paper misses comparison with many subgraph GNNs**, such as Bevilacqua et al 2022, Zhang et al 2023. These methods obtain better performances than MAG-GNN for example on ZINC, using node-based subgraphs (where the bag of subgraph is of size $\vert V \vert$, and therefore are tractable). It is unclear what is the advantage of training an RL agent, which has much higher training time, and using $k$-order policies with $k>1$, if the performances are worse than node-based subgraph GNNs and the inference time is not better.

4. The comparison on the ZINC dataset is unfair because the model does not respect the parameter budget imposed by the dataset (500k). Currently he budget is twice the one that is allowed (three times if we consider the target q-network). Please fix and ensure you remain within the budget.

5. **The paper misses a thorough comparison with the model using the full bag and $k>1$**. Again, it should be based on the same architecture that MAG-GNN uses as downstream prediction model. Only in this way we can know what is the performance achievable with the full bag, and understand how distant MAG-GNN is from it.



Bevilacqua et al 2022. Equivariant Subgraph Aggregation Networks. ICLR 2022

Zhang et al 2023. A Complete Expressiveness Hierarchy for Subgraph GNNs via Subgraph Weisfeiler-Lehman Tests. ICLR 2023

**Questions:**

1. **What is the performance of _your prediction subgraph GNN_ (which is built on k-OSAN) when trained with random subsets of subgraphs (randomly selected at every forward)**? Note that I am asking to keep your downstream prediction subgraph GNN formulation identical to what you use in MAG-GNN (except the hyperparameter choices) in order to truly understand the impact of the RL agent. Please use $m$ and $k$ up to the values you consider for MAG-GNN.
2. Please include comparison with subgraphs GNNs (Bevilacqua et al 2022, Zhang et al 2023) and with k-OSAN with IMLE.
3. Table 7 shows that MAG-GNN with $m=1$ (and ensemble 3) achieves practically the same performance of the MAG-GNN that uses much higher $m$. Please investigate this aspect. Does it imply that MAG-GNN is effectively only learning to select one subgraph, and the other ones are unimportant (as it can get similar performances when choosing one or many)?
4. I think $f_p$ in Equation 9 should be $f_{rl, p}$. Indeed, using $f_p$ means that you take the representation of graph obtained through the prediction subgraph GNN. However, from the code I think you take the representation of graph obtained through the subgraph GNN that parametrizes the Q-network (that is based on $g_{rl}$). Please clarify.
5. In the SIMUL case, do you reset the replay buffer every time you train your downstream prediction subgraph GNN $g_p$?
6. What is the performance of _your prediction subgraph GNN_ when using the full bag and $k>1$ (that is, for various values of $k>1$)?

---

> ### Author Rebuttal · Authors · 2023-08-07
>
> We are very grateful for reviewer jrFK’s in-depth review and insightful comments. We address the reviewer’s concerns and questions as follows:
>
> - W1: Full comparison with a random approach
>
> As mentioned in Q1 in the paper, we also believe that outperforming random subgraphs is critical to the actual utility of the model. We agree with reviewer jkF that a full experiment of MAG-GNN and random approach over $m$ and  $k$ provides a more comprehensive and fair comparison of the two methods. We refer the reviewer to the general response GR1, including the full comparison between MAG-GNN and the random approach.
> The random approach follows reviewer jrFK’s suggestion. We use sum pooling on the GNN graph embeddings of all subgraphs for downstream MLP.
>
> - W2: Comparison to I-MLE
>
> k-OSAN is undeniably a great inspiration for our work. I-MLE also strives for an efficient and effective subgraph selection policy, and it is an important baseline, as we also discussed from lines 236 to 243 and compared to it in Table 6. The key difference between I-MLE and MAG-GNN is that I-MLE is data-driven, while MAG-GNN changes the sampled subgraphs based on high-level graph features. Consequently, I-MLE fails to sample more representative subgraphs when the desired graph feature is beyond the 1-WL test's capability since the model that parametrizes its sampling distribution is an MPNN. Meanwhile, because MAG-GNN operates on labeled graphs which introduces more expressivity to the model, it can change the selected subgraphs according to graph features undetectable by MPNN/I-MLE. In Table 6, Appendix E of our paper, we already compared MAG-GNN’s performance with k-OSAN IMLE’s *best* ZINC performance reported in its paper. However, its performance is even worse than the random approach. Hence, we implemented IMLE using our codebase (adopting IMLE code module to our $f_p$) and reported the results.
>
> |  | ZINC |
> | --- | --- |
> | k-OSAN I-MLE original m=10, k=1 | 0.155 |
> | k-OSAN I-MLE ours m=10, k=1 | 0.121 |
> | Random m=10, k=1 | 0.127 |
> | MAG-GNN m=10, k=1 | 0.110 |
>
> We can see that I-MLE only minimally improves over random while MAG-GNN effectively reduces the test MAE.
>
> - W3: Comparison to node-based GNNs.
>
> We agree that comparing the state-of-the-art node-based methods is important, and we refer the reviewer to the general response GR2 for their comparison to MAG-GNN.
>
> - W4: Fair ZINC comparison.
>
> Following the reviewer’s comments, we conducted all ZINC follow-up experiments using a hidden dimension of 64 (previously 100) to respect the 500K parameter constraint.
>
> - W5: Full bag experiments.
>
> We agree with the reviewer the full bag setting is a meaningful comparison to MAG-GNN. Such a setting implies that a GNN is applied to all k-node-rooted subgraphs, and a pooling function is applied to get the final embedding. Unfortunately, this is generally intractable to train for real-world datasets, even for k>1. In fact, this drawback motivates many works trying to realize k-WL expressivity using less computation resources, including our work. Under our specific implementation, using a full bag is essentially a k-OSAN algorithm, as we mentioned in the preliminary. **Note that the original k-OSAN paper did not include full bag experiments (only k-OSAN with I-MLE reported), most likely due to its intractability.** Note that the k-OSAN result we reported in table 6 is the best performance reported in the original k-OSAN paper with I-MLE. Nevertheless, we still managed to perform $k\leq 2$ full bag training on SR25 and ZINC datasets.
>
> |  | ZINC | SR25 |
> | --- | --- | --- |
> | k=1 | 0.118 | 6.7 |
> | k=2 | 0.261 | 100 |
> | 3-WLGNN | 0.256 | 6.7 |
> | MAG-GNN | 0.104 | 100 |
>
> We can see that MPNN even outperforms k=2 full bag training on ZINC dataset. This phenomenon aligns with 3-WLGNN results [1], which also realizes high-dimensional WL with neural networks. While they are theoretically more powerful, properly implementing them on real-world datasets is very challenging, potentially due to the implied larger hidden size required for injective mapping.
>
> - Q3: Ensemble and m>1
>
> We want to clarify that the result that the reviewer mentioned is observed when we train a MAG-GNN that performs 6 RL steps, and we take the last generated node tuple. The ensemble is on the node tuples over timesteps but not m. However, the same best performance (without pre-training) is obtained with only two RL steps but more node tuples (faster inference). We would also like to highlight the best ZINC performance with pre-training, obtained when m>1, meaning that m>1 is still essential for a more comprehensive graph representation. Here we present a list of $m$ used during hyper-parameter searching, and mark chosen $m$ in **bold**
>
> |  | m |
> | --- | --- |
> | ZINC | {2,**3**,4} |
> | ZINC-FULL | {2,**3**,4} |
> | CYCLES | {2,3,**4**} |
> | MOLHIV | {**2**,3,4} |
> | EXP | {**1**,2,3} |
> | SR25 | {1,2,**3**} |
> | CSL | {1,**2**,3} |
>
> - Q4: Reviewer is jrFK is correct on this notation problem. We will reflect on and correct this in our revision.
> - Q5: We do not reset the buffer every time we update the prediction GNN. As mentioned in Section 4, this causes inconsistency during training, but the models gradually become stable.
>
> [1] Morris, Christopher, et al. "Weisfeiler and leman go neural: Higher-order graph neural networks." *Proceedings of the AAAI conference on artificial intelligence*. Vol. 33. No. 01. 2019.

---

> ### Author Response · Authors · 2023-08-14
>
> Dear Reviewer jrFK,
>
> Again, we sincerely thank reviewer jfFK for the time and effort spent on the detailed comments. Many points you raised inspired us to evaluate our methods comprehensively. We just want to reach out to you at the end of the discussion session and see if our rebuttal addresses your main concerns. In short, we compare the random approach on more datasets varying $m$ and $k$, compare MAG-GNN with SOTA node-based subgraph GNN, and explain why full bag training was not included originally. We agree these are critical to present our work and are happy to include them in our revision to enhance the work. Please let us know any further questions you have, and we look forward to the discussion with you.
>
> Submission 3696 authors

---

> > ### Comment · Area_Chair_PWZo · 2023-08-16
> > **Rebuttal to Reviewer jrFK**
> >
> > Reviewer jrFK, this seems like a thoughtful review. Did the rebuttal address your concerns? Thanks!

---

> > > ### Comment · Reviewer_jrFK · 2023-08-16
> > >
> > > I appreciate the author's effort in the rebuttal, especially in adding the comparison to the random baseline.
> > > However, I would like to further raise a few points of discussion.
> > >
> > > 1. *Is the comparison with random fair in terms of number of parameters?* As discussed in the review, in the ZINC dataset your model was out of budget. Using a random policy reduces the number of parameters by 1/3. Since models on ZINC are known to have better performances with larger number of parameters, have you considered the impact of this? Generally speaking, I think you should report the number of parameters both without including the target q-network and when including it. In your reply you said you would run with hidden size 64 to respect the 500K budget, but results are not reported.
> > > 2. Can you try using mean pooling instead of sum pooling for the full bag training when k=2? I think the poor results you obtain with the full policy (similar to a MPNN) might be due to the aggregation that explodes when the number of subgraphs is too large.
> > > 3. Since your model is slightly different than k-OSAN (because you get different results), what are the results of the full bag when k=1?

---

> > > > ### Author Response · Authors · 2023-08-17
> > > >
> > > > We thank Reviewer jrFK for the continued discussion, and address the questions as follows.
> > > >
> > > > - Is the comparison with random fair in terms of number of parameters? As discussed in the review, in the ZINC dataset your model was out of budget. Using a random policy reduces the number of parameters by 1/3. Since models on ZINC are known to have better performances with larger number of parameters, have you considered the impact of this? Generally speaking, I think you should report the number of parameters both without including the target q-network and when including it. In your reply you said you would run with hidden size 64 to respect the 500K budget, but results are not reported.
> > > >
> > > > We strongly agree with the reviewer that MAG-GNN should stay within the parameters budget, and we want to clarify that **all ZINC experiments in the rebuttal have <500k parameters using a hidden dimension of 64**. Specifically, the number of parameters in the q-network is 202k, the prediction GNN is 253k, which add up to 455k<500k, and the number of target q-network is the same as the q-network 202k. However, because we do not directly train the target q-network (only update from the q-network) and we do not need the target network during inference (only 455k parameters used during inference), we believe this is still a fair setting. Comparing the ZINC experiment results presented in the rebuttal (<500k parameters) and the experiment in the paper, we found that MAG-GNN seems to be less sensitive to the hidden dimension, possibly because the q-network filtered out non-informative subgraphs, and the prediction network only needs to remember the information of a smaller set of subgraphs. For the random approach, **we stay with the 100 hidden dimension size**, resulting in 488k parameters. We will add the parameters comparison in our revision.
> > > >
> > > > - Can you try using mean pooling instead of sum pooling for the full bag training when k=2? I think the poor results you obtain with the full policy (similar to a MPNN) might be due to the aggregation that explodes when the number of subgraphs is too large.
> > > >
> > > > We would like first to bring up the point that we use batch normalization in our task MLP, which should also alleviate the exploding subgraphs issue. Nevertheless, we conducted k=2 full bag training using mean pooling.
> > > >
> > > > |  | ZINC | SR25 |
> > > > | --- | :---: | :---: |
> > > > | k=1 (SUM) | 0.118 | 6.7 |
> > > > | k=2 (SUM) | 0.261 | 100 |
> > > > | k=2 (MEAN) | 0.247 | 100 |
> > > > | 3-WLGNN | 0.256 | 6.7 |
> > > > | MAG-GNN | 0.104 | 100 |
> > > >
> > > > Mean pooling seems to have slightly better performance but is still sub-optimal, possibly because mean pooling is not as injective as sum pooling, which makes it difficult for the MLP to distinguish graphs.
> > > >
> > > > - Since your model is slightly different than k-OSAN (because you get different results), what are the results of the full bag when k=1?
> > > >
> > > > The table in the response of W5 in the original rebuttal includes k=1 full bag results, and we paste the results in the table to the last question.

---

> > > > > ### Comment · Reviewer_jrFK · 2023-08-18
> > > > >
> > > > > I think overall this paper represents a valuable contribution to the community.
> > > > > I have increased my score, but for the next revision I urge the authors to provide the comparison with the random baseline **in each table**.

---

> > > > > > ### Author Response · Authors · 2023-08-19
> > > > > >
> > > > > > We appreciate the reviewer for the initial thoughtful comments and responsible follow-ups. We will revise according to our discussion and thank you for your time and effort helping us improve the paper.

---

### Official Review · Reviewer_SjU6 · 2023-07-06

**Soundness:** 3 good
**Presentation:** 3 good
**Contribution:** 2 fair
**Rating:** 6
**Confidence:** 2

**Summary:**

The paper aims to locate expressive subgraphs for graph representation learning. To do so, the author proposed to use deep Q Learning for training an efficient agent which optimizes the combinatorial problem of choosing the optimal subgraph. The proposed agent-aided GNN shows good performance and is able to transfer to real-world datasets.

**Strengths:**

Motivation is interesting and the methodology is well explained. The experiment result seems sound. The proposed model is on par with state-of-the-art while using less computational resources.

**Weaknesses:**

* How to train an effective agent seems to be a critical point. Despite that, the author has a brief discussion in sec 4.1, more quantitative analysis and illustration on this part can better support the author's choice of ORD.
* The proposed method seems to be on par with the state-of-the-art. While I don't think absolute performance is an important concern given the method is more efficient. I think it's important to investigate in L316 Q3 how much of the RL agent expressiveness benefit the real-world datasets compared to random ones or human-designed ones.

-------------------------------------------------------------------------------------
I've read the author's response to me and other reviewers which resolves my concern. I would like to remain my rating as weak accept.

**Questions:**

* In table4, how is no RPE evaluated, why is no RPE on par with training agent on 3-CYCLE 6-CYCLE and SR25?

---

> ### Author Rebuttal · Authors · 2023-08-07
>
> We thank Reviewer SjU6 for acknowledging our contribution and appreciate the reviewer's suggestions to improve our work. We address the reviewer's concerns and questions as follows,
>
> - W1: How to train an effective agent seems to be a critical point. Despite that, the author has a brief discussion in sec 4.1, more quantitative analysis and illustration on this part can better support the author's choice of ORD.
>
> We want to clarify that we use ORD and SIMUL training paradigms in different scenarios, as described in the experiment section. We also agree with Reviewer SjU6 that a quantitative study of the effect of different training paradigms is beneficial, and we present the model’s performance on ZINC and SR25 with ORD and SIMUL paradigms (PRE is not considered here because the agent is not trained during fine-tuning stage).
>
> |  | ZINC | SR25 |
> | --- | --- | --- |
> | SIMUL | 0.104 | 100 |
> | ORD | 0.131 | 100 |
>
> We also present the training time on these scenarios:
>
> |  | ZINC | SR25 |
> | --- | --- | --- |
> | SIMUL | 182min | 204min |
> | ORD | 109min | 141min |
>
> ORD usually leads to faster convergence because of its stability. However, because the prediction GNN could be ignorant of the node labels if trained without the agent, as discussed in Section 4.1, the agent trained in the ORD paradigm sometimes only minimally improves the prediction GNN, leading to worse performance than SIMUL.
>
> - W2: The proposed method seems to be on par with the state-of-the-art. While I don't think absolute performance is an important concern given the method is more efficient. I think it's important to investigate in L316 Q3 how much of the RL agent expressiveness benefit the real-world datasets compared to random ones or human-designed ones.
>
> We thank reviewer SjU6 for emphasizing the comparison between MAG-GNN and the random approach, and we believe such evaluation is essential to understand the realistic utility of the model. We refer the reviewer to the general response GR1 where we provide a full comparison between MAG-GNN and random approach varying m and k on real-world datasets.
>
> We show that MAG-GNN outperforms the random approach under all circumstances, meaning that the node tuples that MAG-GNN finds are more discriminative.
>
> - Q1: In table4, how is no RPE evaluated, why is no RPE on par with training agent on 3-CYCLE 6-CYCLE and SR25?
>
> NO PRE is evaluated without any pretraining on the synthetic datasets. It is essentially the SIMUL training paradigm, and we use NO PRE to emphasize the comparison. NO PRE achieves close performance because the underlying architectures of the NO PRE and pre-trained models are the same. They differ only in the parameter values, meaning a model without pretraining can potentially achieve the same performance. The PRE model aims to help the RL agent recognize better high-level patterns in a noise-free environment, such as the synthetic high expressivity datasets. Meanwhile, the numbers might be small, but the MSE reduction for ZINC and ZINC-FULL is 9.43% and 23.3%, respectively, which is a considerable performance improvement, showing that the PRE paradigm helps generalize high-level graph features to downstream tasks.

---

> ### Author Response · Authors · 2023-08-14
>
> Dear Reviewer SjU6,
>
> Thank you again for acknowledging our contribution. We just want to check in and see if our rebuttal addressed the weaknesses you mentioned at the end of the reviewer period. In the rebuttal, we provide a timetable to discuss the difference between ORD and SIMUL training and clarify the PRE experiment setup. We also offer a more comprehensive comparison to the random approach. We really appreciate your critical comments encouraging us to inspect our approach further, and look forward to your reply.
>
> Submission 3693 authors

---

### Official Review · Reviewer_kfK6 · 2023-07-16

**Soundness:** 3 good
**Presentation:** 3 good
**Contribution:** 3 good
**Rating:** 7
**Confidence:** 2

**Summary:**

this paper proposes a strategy to select the optimal subset instead of extensive enumeration for the subgraph GNN. The proposed method can achieve comparative results and reduce the running time.

**Strengths:**

1. the paper is well written and easy to follow.
2. the experiment part is thoroughly detailed。
3. the experiment shows the model can achieve the comparative result with subgraph GNN while significantly reducing the running time.

**Weaknesses:**

N/A

**Questions:**

N/A

---

> ### Author Rebuttal · Authors · 2023-08-07
>
> We thank the reviewer for the positive feedback and refer the reviewer to our other responses in case the reviewer is interested in more details about our work. We are also happy to answer any further questions the reviewer might have.

---

### Official Review · Reviewer_FiDZ · 2023-07-25

**Soundness:** 2 fair
**Presentation:** 3 good
**Contribution:** 2 fair
**Rating:** 4
**Confidence:** 3

**Summary:**

This paper proposes a reinforcement learning boosted approach to subgraph graph neural networks (GNNs) that achieves high expressivity with constant complexity, instead of the exponential complexity of existing methods. They demonstrate the effectiveness and efficiency of their approach through experiments that compare it with the state-of-the-art methods.

**Strengths:**

1. The paper is well-written.
2. The proposed MAG-GNN can achieve good expressivity while reducing the time complexity of subgraph enumeration.


**Weaknesses:**

1. The authors report that most methods in Table 1 achieve 100% accuracy on the dataset. This suggests that the dataset may be too simple to adequately evaluate the expressivity of a model. The authors should consider applying more challenging datasets to better assess the performance of their models.
2. The table captions are vague and do not provide sufficient information for readers to interpret the experimental results properly.
3. In Section 3, the authors use the same data for training and test, which may lead to overfitting and unrealistic performance.
4. The notation |V^2| should be |V|^2.
5. The proof of the theorem is unclear and lacks details.


**Questions:**

It seems no supporting for this statement: This reduces the exponential complexity of subgraph enumeration to the constant complexity of a subgraph search algorithm while keeping good expressivity

**Limitations:**

The paper does not discuss limitations or potential social impacts.

---

> ### Author Rebuttal · Authors · 2023-08-07
>
> We appreciate the feedback from reviewer FiDZ, and would like to introduce our research's background and intuition further. In traditional deep-learning fields (e.g. CV and MLP), expressivity usually is not an issue, and reducing overfitting is more critical. However, in GNN, the expressivity is limited by 1-WL test [1], preventing the model from distinguishing many different graphs. Hence, a downstream predictor cannot correctly classify these graphs regardless of how the predictor generalizes. Subgraph GNN rises as a solution to the limited expressivity issue, and we observe that as the expressivity increases, the test performance on real-world datasets also increases. However, the drawback of subgraph GNN is its worse runtime because it needs to extract multiple subgraphs to run GNN multiple times. This motivates MAG-GNN that finds the most informative subgraphs to reduce runtime while keeping subgraph GNN's expressivity. We also address the reviewer's questions and concerns as follows,
>
> - W1: Are 100% accuracy datasets too simple?
>
> We agree that a more challenging dataset will provide a meaningful evaluation of our model. However, since a widely-acknowledged expressivity dataset is missing from the literature, we follow the convention [2,3,4] that evaluates the model on these synthetic datasets to validate the model’s claimed expressivity. Generally, models expressive enough will have a perfect prediction on these datasets, while models with insufficient expressivity will collapse to random guesses, like GIN on the EXP dataset and NGNN on the SR25 dataset. We also provide an evaluation of other real-world datasets, including ZINC, OGBG-MOLHIV, etc., in Tables 2, 3, and 6 to comprehensively evaluate our model.
>
> - W2: Vague table captions.
>
> Due to the space limitation, we keep the table description succinct and describe the experiments in the experimental sections. When more space is permitted in the revision, we will add more detailed descriptions to the table captions.
>
> - W3: In Section 3, the authors use the same data for training and test, which may lead to overfitting and unrealistic performance.
>
> We follow the convention to train and evaluate the model on the same SR25 dataset [2,4]. The evaluation aims to examine the expressivity instead of the generalizability. Hence the experiment protocol tests if a model can discriminate every graph in the dataset rather than test if knowledge learned from one dataset can be generalized to another.  As the background introduction mentioned, expressivity is particularly important in GNN research, and this motivation section is to show only a few subgraphs achieve similar expressivity. Meanwhile, MAG-GNN’s ability to generalize can be seen in its comparable performance to state-of-the-art methods in real-world datasets.
>
> - W4: The notation |V^2| should be |V|^2.
>
> We appreciate the reviewer’s careful reading of the paper. Here, we use the power of the set of nodes, $V$, to represent the node tuple set as a convention widely adopted in the literature. For example, $V^2$ means the set of all 2-node-tuples, and $|V^2|$ means the cardinality of such set, while $|V|^2$ is the square of the cardinality of the set of nodes. While both numbers are the same, we believe the former might be more appropriate to our context. Nevertheless, we will explain this in detail in our revision.
>
> - W5: The proof of the theorem is unclear and lacks details.
>
> We thank the reviewer’s in-depth reading into the proof. We will clarify assumptions and add more details to the proof. Meanwhile, would you kindly let us know which part of the proof causes confusion and we are more than happy to clarify. For readers to understand the proof better, we present the intuition of the proofs.
>
> Theorem 1 is based on the lemma from NGNN [3] that the representation of a subgraph in $G_1$ generated from a GNN is very likely to be different from that of another subgraph in $G_2$. We can then use GNN with more layers to simulate a BFS and the GNN in the lemma. Lastly, we use a union bound to complete the proof.
>
> Theorem 2 constructs super-graphs by two base CSL graphs that 1-node-marked subgraphs cannot distinguish. Hence, the super-graphs’ classes can only be determined when the 2-node-marks land on the opposite side of the super-graph. This scenario can be quickly learned by MAG-GNN but not by random approaches, which proves the theorem.
>
> Theorem 3 uses the fact that as long as we correctly label the order and indices of the selected node-tuples, we can injectively map the output of RL GNN to PF-GNN’s output. We construct a GNN with sufficient layers to realize the mapping.
>
> - Q1: It seems no supporting for this statement: This reduces the exponential complexity of subgraph enumeration to the constant complexity of a subgraph search algorithm while keeping good expressivity
>
> This is, in fact, the paper's main contribution and can be seen by the better complexity and competitive performance of our model. Specifically, to achieve k-WL expressivity, the number of GNN runs required for subgraph methods is $O(|V|^k)$, which grows exponentially with k. In contrast, MAG-GNN achieves very competitive performance with $O(mt|V|^2)$ (Here $O(|V|^2)$ is the complexity for running MPNN on a dense graph, $mt$ is the number of MPNN runs. See Appendix D for more details), which does not grow exponentially with k. This, in turn, supports the statement.
>
> [1] Xu, Keyulu, et al. "How powerful are graph neural networks?." arXiv preprint arXiv:1810.00826 (2018).
>
> [2] Feng, Jiarui, et al. "How powerful are k-hop message passing graph neural networks." *Advances in Neural Information Processing Systems* 35 (2022): 4776-4790.
>
> [3] Zhang, Muhan, and Pan Li. "Nested graph neural networks." *Advances in Neural Information Processing Systems* 34 (2021): 15734-15747.
>
> [4] Huang, Yinan, et al. "Boosting the Cycle Counting Power of Graph Neural Networks with I $^ 2$-GNNs." *arXiv preprint arXiv:2210.13978* (2022).

---

> ### Author Response · Authors · 2023-08-14
>
> Dear Reviewer FiDZ,
>
> Thank you again for acknowledging our contribution and raising valuable points to improve our paper. As the discussion period ends soon, we are just wondering if our rebuttal addresses your concerns. Following your comments, we explain the usage of datasets and clarify the notion and the intuition behind our theory. We provided more background to our work to highlight further our contribution, which we plan to include in our revision. We hope this helps present our paper in a clearer way to you.
>
> Looking forward to your reply!
>
> Submission 3693 authors

---

### Official Review · Reviewer_egpV · 2023-07-25

**Soundness:** 3 good
**Presentation:** 3 good
**Contribution:** 3 good
**Rating:** 7
**Confidence:** 3

**Summary:**

The paper introduces MAG-GNN, an RL approach for identifying important subgraphs. This is valuable since models using fewer subgraphs incur lower inference costs. Experimentally, the authors show that MAG-GNN retains the strong performance of sub-graph approaches while reducing the run time.

**Strengths:**

The guiding question of the paper "Are a small number of subgraphs sufficient for graph problems?" is well-motivated. The proposed RL-based approach for identifying subgraphs is well-explained. The approach's performance and efficiency are validated on various benchmark datasets. The negative results of the node-level task are helpful, revealing both the current limit and future potential of the paper.


**Weaknesses:**

**Training efficiency**. While MAG-GNN accelerates inference, the training process becomes more time-consuming. On smaller graphs, this training pipeline may dominate the overall computational costs.

**Task scope**. MAG-GNN excels in graph-level tasks but underperforms in node-level tasks, as the authors gracefully acknowledge.

**Baseline comparison**. The authors primarily compare MAG-GNN with methods using full subgraphs in the experiment section. Considering its efficiency goals, comparing it with other efficient alternatives, such as applying node/edge pruning approaches to the full graph and then using standard subgraph methods, would be useful.

**Questions:**


Could the authors report the training time of MAG-GNN? Knowing the timeframe compared to baselines would be helpful, given that the training of MAG-GNN is costly.

I am wondering if it is possible to visualize the discarded and remaining subgraph structures resulted from RL. Given the complexity and freedom of tuning of the reported RL approach, such visualizations could aid in understanding which subgraph structures are useful and which are not. In particular, it will be informative to apply this visualization technique to toy-like graphs, similar to those in Figure 1, where we already know which subgraph structures matter.

**Limitations:**

See the "Weaknesses" section above.

---

> ### Author Rebuttal · Authors · 2023-08-03
>
> We thank reviewer egpV for the insightful review, and we address reviewer egpV’s questions and concerns as follows,
>
> - W1: **Training efficiency**. While MAG-GNN accelerates inference, the training process becomes more time-consuming. On smaller graphs, this training pipeline may dominate the overall computational costs.
>
> Reviewer egpV is keen to observe that the critical trade-off in MAG-GNN is between training and inference times. MAG-GNN emphasizes its optimal **inference complexity**, which is **more important** practically, as once trained, the model can be reused on potentially infinite data. The training time for MAG-GNN is less correlated to the graph size, as the training step complexity is linear to the graph size but more correlated to the inherent property of the graphs and the prediction targets. We present the convergence time to obtain the reported performance on ZINC and CYCLE-6 datasets of similar graph sizes.
>
> |  | ZINC | CYCLE-6 |
> | --- | :---: |:---: |
> | NGNN | 36min | 14min |
> | I^2GNN | 127min | 27min |
> | MAG-GNN | 182min | 156min |
>
> NGNN converges much faster than I^2GNN and MAG-GNN, while MAG-GNN has better performance than NGNN on both datasets (see Tables 2 and 4 in the main paper). MAG-GNN is roughly 1.5 times slower in training compared to I^2GNN but much faster for inference, as shown in Table 5 of the main the paper. We note that training MAG-GNN on CYCLE-6 is slower, potentially because the cycle-6 task creates a larger search space for MAG-GNN to find appropriate policies.
>
> We also want to point out that the training time of the MAG-GNN can be significantly reduced when we use the PRE training scheme, where we first train on a synthetic expressivity dataset and transfer the knowledge to a real-world dataset. After pretraining on CYCLE-6, the convergence time of training the prediction network is **32min** which is even faster than that of NGNN. This shows that the trained agent can be reused and efficiently generalized to downstream real-world tasks, alleviating the high training cost problem.
>
> - W2: **Task scope**. MAG-GNN excels in graph-level tasks but underperforms in node-level tasks, as the authors gracefully acknowledge.
>
> As the reviewer points out, improving node-level performance is a valuable future direction. A potential solution is to design a better node-level reward function. Meanwhile, we note that the inference time of MAG-GNN on these tasks is more optimal compared to methods that achieve better performance, and effort spent on adapting MAG-GNN to node-level tasks can lead to effective and efficient method.
>
> - W3: **Baseline comparison**. The authors primarily compare MAG-GNN with methods using full subgraphs in the experiment section. Considering its efficiency goals, comparing it with other efficient alternatives, such as applying node/edge pruning approaches to the full graph and then using standard subgraph methods, would be useful.
>
> We agree such a pruning+subgraph GNN method approach can be an interesting comparison. We did some literature search and didn’t find such existing approaches. Instead,  we provide results of the following experiment where we apply random edge drop with a drop probability p on every edge and then apply NGNN onto edge dropped graphs.
>
> |  | ZINC | Inference Time on 10 tests (ms) |
> | --- | :---: | :---: |
> | p=0.2 | 0.164 | 152.7 |
> | p=0.4 | 0.274 | 137.0 |
> | MAG-GNN | 0.104 | 159.1 |
>
> The experiments first show that, indeed, by pruning the subgraphs, we can reduce the inference of subgraph GNN. However, we also notice the sharp performance drop of NGNN applied to pruned graphs, meaning that the full graph structure is still critical to optimal performance. MAG-GNN reduces running time while effectively utilizing the full graph structure.
>
> - Q2: I am wondering if it is possible to visualize the discarded and remaining subgraph structures resulted from RL. Given the complexity and freedom of tuning of the reported RL approach, such visualizations could aid in understanding which subgraph structures are useful and which are not. In particular, it will be informative to apply this visualization technique to toy-like graphs, similar to those in Figure 1, where we already know which subgraph structures matter.
>
> We agree with the reviewer and believe a visualization will be beneficial in understanding the mechanism of the methods. We present two examples in Figures 1 and 2 in the general response's pdf. One example is in EXP dataset using a 2-node-mark. Most graphs in the EXP dataset have two components. MAG-GNN’s most prevalent pattern is that whenever the initial random node markings land on the same component, the agent will move one marking to the other, identifying the graph structure. This is not possible if the node markings only land on one component.
>
> The other interesting example is when we train on the toy example to distinguish the Shrikhande graph and the Rook 4-by-4 graph. The agent always produces 2-node-markings that are 2-distance away on both graphs. Moreover, the agent always chooses to move one node marking of the Rook 4-by-4 graph to its neighbor, while it chooses to be stagnant in the Shrikhande graph when the common neighbors of the two marked nodes have an edge in between.

---

> > ### Comment · Reviewer_egpV · 2023-08-14
> > **After rebuttal**
> >
> > Thanks to the authors for their response. Their reply addressed my questions, and I raised my score accordingly. I think it would be interesting to compare non-random edge drop experiments, though, in a context similar to the lottery ticket hypothesis for graphs (search SNIP, Synflow, etc for attempts on tabular and imagery data). Perhaps this is something worth exploring, if the authors have the time and interest.

---

> > > ### Author Response · Authors · 2023-08-14
> > > **Thank you**
> > >
> > > Thank you very much for reading our response and raising your score. We sincerely appreciate the proposed idea and would love to inspect it further.

---

> ### Author Response · Authors · 2023-08-14
>
> Dear Reviewer egpV,
>
> As the discussion period ends soon, we just want to reach out and see if our rebuttal answers your questions. In short, we present a detailed training time comparison, a performance comparison to the method you suggested, and visualizations of the agent steps in the attached pdf to address your concerns. Thank you again for your comments and suggestions to improve our paper! We look forward to your reply.
>
> Submission 3693 authors

---

### Author Rebuttal · Authors · 2023-08-07

We thank all reviewers for their constructive comments that help us improve our work. Here we address some common questions and concerns that reviewers raise, and we also respond to their comments individually.

- GR1: MAG-GNN is a random method that uses an RL agent to refine the selected subgraph, a full comparison with a pure random approach is required to show the necessity and utility of such agent.

This is also the question that we try to answer in Q1 in the experiment section, and following reviewer SjU6 and jrFK’s suggestion, we also believe a more comprehensive study showing that MAG-GNN outperforms random subgraphs is critical to the actual utility of the model. We compare MAG-GNN and the random subgraph approach over $m$ and  $k$ on real-world datasets.

The random approach follows reviewer jrFK’s suggestion. We use sum pooling on the GNN graph embeddings of all subgraphs for a fair comparison. The results to the left of the bar in a cell are the results of the random approach, and the results to the left are MAG-GNN results.

| SR25 |  | m=1 |  | m=2 |  | m=3 |
| --- | --- | --- | --- | --- | --- | --- |
| k=1 |  | 6.7  \|  6.7 |  | 6.7   \|   6.7 |  | 6.7  \|  6.7 |
| k=2 |  | 65.4    \|    100 |  | 82.3    \|    100 |  | 89.6    \|   100 |
| k=3 |  | 67.4    \|     100 |  | 83.9    \|    100 |  | 93.8    \|    100 |

Compared to random approach that does not achieve that perfect score even when m=3 and k=3, MAG-GNN achieves perfect score with k=2 and m=1, meaning that MAG-GNN effectively finds more representative node-tuples.

We conducted a similar comparison on the ZINC dataset.

| ZINC |  | m=2 |  | m=3 |  | m=4 |
| --- | --- | --- | --- | --- | --- | --- |
| k=3 |  | 0.131  \|  0.115 |  | 0.124   \|   0.111 |  | 0.125  \|  0.110 |
| k=4 |  | 0.135    \|    0.106 |  | 0.131   \|    0.104 |  | 0.134    \|   0.101 |
| k=5 |  | 0.132    \|     0.112 |  | 0.136    \|    0.114 |  | 0.131    \|    0.109 |
| k=6 |  | 0.144    \|     0.124 |  | 0.140    \|     0.127 |  | 0.136   \|    0.116 |

The best validation score for MAG-GNN is obtained at m=3, k=4, leading to a test score of 0.104. We note that MAG-GNN outperforms the random approach under all configurations, showing that MAG-GNN also performs better on real-world datasets.

A similar pattern can be found in the OGBG-MOLHIV dataset:

| MOLHIV | | m=2 | | m=3 | | m=4 |
| ---|--- | --- | --- | --- |---|---|
| k=3| | 75.21  \|  77.12 || 76.06   \|   77.26 || 76.29  \|  76.52 |
| k=4 || 74.69   \|   76.53 || 75.92   \|    78.10 || 76.51    \|   76.28 |
| k=5 || 75.17    \|     74.95 || 74.83    \|    76.81 || 75.37    \|    76.30 |
| k=6 || 74.28    \|     74.64 || 75.20    \|     76.14 || 75.68   \|   76.19 |

MAG-GNN also outperforms the random approach in most cases.

Because the QM9 dataset has many labels, due to the space limitation, we pick $U_0$ where usually higher expressivity GNN achieves a better score:

| QM9-$U_0$ |  | m=2 |  | m=3 |  | m=4 |
| --- | --- | --- | --- | --- | --- | --- |
| k=3 |  | 0.306 \|  0.241 |  | 0.318   \|   0.205 |  | 0.285  \|  0.153 |
| k=4 |  | 0.327    \|    0.144 |  | 0.309   \|    0.111 |  | 0.291    \|   0.119 |
| k=5 |  | 0.297    \|     0.137 |  | 0.288    \|    0.125 |  | 0.294    \|    0.121 |
| k=6 |  | 0.336    \|     0.168 |  | 0.314    \|     0.154 |  | 0.296   \|    0.146 |

QM9-$U_0$ is more sensitive to the hyperparameters. We can still see that the random approach brings minimal performance as $m$ increases, while MAG-GNN obtains good performance with appropriate m and k.

- GR2: While MAG-GNN approximates the expressivity of intractable higher-order WL, how does MAG-GNN compare to tractable note-based methods?

We agree that comparing the state-of-the-art node-based methods is important, and we present MAG-GNN's results against them on ZINC, ZINC-FULL and SR25 datasets.

|  | ZINC | ZINC-FULL | SR25 |
| --- | --- | --- | --- |
| SUN[1] | 0.083 | 0.024 | 6.7 |
| SSWL+ [2] | 0.070 | 0.022 | 6.7 |
| MAG-GNN | 0.104 | 0.029 | 100 |
| MAG-GNN PRE | 0.098 | 0.023 | - |

State-of-the-art node subgraph methods on the ZINC dataset outperform MAG-GNN. However, it is also important to note that they do not acquire >3-WL expressivity (random guesses on the SR25 dataset) as MAG-GNN does, meaning that they will potentially fail on datasets requiring higher expressivity. The higher expressivity of MAG-GNN requires more data. We also compared pre-trained MAG-GNN with these models. We find that after pre-training, MAG-GNN is very close to these models on ZINC-FULL when more data is available. We also would like to emphasize that MAG-GNN is more optimal in terms of inference time and show an inference time comparison between MAG-GNN and these models.

| time (ms) | ZINC-FULL | CYCLE |
| --- | --- | --- |
| SUN | 2590.2 | 1410.5 |
| SSWL+ | 2682.6 | 1473.8 |
| MAG-GNN m=3 k=2 | 321.6 | 155.1 |

Note that MAG-GNN, even when k=2, feeds much fewer subgraphs to the MPNN compared to SSWL+ and SUN, which results in better inference time. MAG-GNN is designed to reduce the $O(n^k)$ subgraphs for $k$-tuple-based subgraph GNNs to a constant while still maintaining the maximum potential expressive power the subgraph GNNs can achieve.

- GR3: More details on different training paradigms.

Per reviewer 9Mgd's request, we added pseudocode of our different training paradigms in the attached pdf.

[1] Frasca, Fabrizio, et al. "Understanding and extending subgraph gnns by rethinking their symmetries." Advances in Neural Information Processing Systems 35 (2022): 31376-31390.

[2] Zhang, Bohang, et al. "Rethinking the expressive power of gnns via graph biconnectivity." arXiv preprint arXiv:2301.09505 (2023).

---

### Decision · Program_Chairs · 2023-09-21

**Decision:**

Accept (poster)

**Comment:**

This paper proposes a reinforcement learning boosted approach to subgraph graph neural networks (GNNs). This addresses the exponential complexity of exact subgraph GNNs in comparison to regular MPNNs (since they require to enumerate all subgraphs of a graph). The authors evaluated the need of this exhaustive enumeration, finding that often a select subset of subgraphs offers the same discriminative ability as all subgraphs. Inspired by this observation, an RL-based approach (MAG-GNN) is introduced by the authors to determine the most effective subset of subgraphs for the specific task being addressed.

After reading the paper, all reviews, and rebuttal, I believe there is reasonable consensus that the original submission should have considered a few other scenarios (e.g., random policy, better comparison with existing work), as aptly pointed out by reviewer jrFK (which provided a high-quality review and was highly engaged during the rebuttal). The rebuttal, however, seems to have answered most of these concerns. I am happy to recommend acceptance.